# Age-related constraints on the spatial geometry of the brain

Yuritza Y. Escalante [1,5], Jenna N. Adams[1,5], Michael A. Yassa [1] &
Niels Janssen [2,3,4] ✉

Age-related structural brain changes may be better captured by assessing complex spatial geometric differences rather than isolated changes to individual regions. We applied an analytic method to quantify age-related changes to the spatial anatomy of the brain by measuring expansion and compression of global brain shape and the distance between cross-hemisphere homologous regions. To test how global brain shape and regional distances are affected by aging, we analyzed 2603 structural MRIs (range: 30−97 years). Increasing age was associated with global expansion across inferior-anterior gradients, global compression across superior-posterior gradients, and regional expansion between frontotemporal homologues. Specific patterns of global and regional expansion and compression were further associated with clinical impairment and distinctly related to deficits in various cognitive domains. These findings suggest that changes to the complex spatial anatomy and geometry of the aging brain may be associated with reduced efficiency and cognitive dysfunction in older adults.

The brain undergoes major structural changes during aging. Changes in brain volume follow an inverted U-shape trajectory as people age, with increases in volume across development, relative stability in adulthood, and then decreases due to neurodegeneration in old age[1–4]. Gross anatomical changes such as atrophy, sulcal widening, and ventricular enlargement are minimal before age 50, and then increase dramatically in the years that follow[5–8]. Further, there is a decrease in overall grey matter and white matter beginning after midlife[9–11], with atrophy most pronounced in the frontal, parietal, and temporal lobes, suggesting regional vulnerabilities within the aging process.

Structural changes in the human brain are primarily assessed using T1-weighted magnetic resonance imaging (MRI) scans, applying varying methods of analysis. Previous studies have focused on assessing isolated changes to regional volumes, cortical thickness, or voxelwise deformation patterns using template methods such as voxel-based morphometry. Volumetric analyses have demonstrated that the majority of brain regions decrease in volume with older age, with

anterior prefrontal, hippocampal, caudate, and posterior parietal regions of the brain being most strongly affected[12–15], while voxel-based morphometry also implicates age-related decreases in prefrontal cortex[16,17]. Collectively, these studies substantiated that aging has widespread effects on both cortical and subcortical structure, though particular regions implicated tend to vary across studies.

While these previous methods account for how the volume of each region or deformation of each voxel changes in *isolation* in respect to aging, these standard approaches are not designed to characterize complex, simultaneous changes in the spatial relationships between regions. However, understanding how aging affects geometric features such as the global shape of the brain or spatial distances between regions would provide critical insight into the overall anatomical reorganization of the brain with age. This is particularly important to characterize because changes to the complex spatial anatomy and geometry of the brain may impact the ability of the brain to properly function and process information[18,19]. Recent work has shown that the shape of the

[1]Department of Neurobiology and Behavior and Center for the Neurobiology of Learning and Memory, University of California, Irvine, CA, USA. [2]Instituto Universitario de Neurociencia (IUNE), Universidad de La Laguna, Santa Cruz de Tenerife, Spain. [3]Instituto Universitario de Tecnologías Biomédicas (ITB), Universidad de La Laguna, Santa Cruz de Tenerife, Spain. [4]Department of Psychology, Universidad de La Laguna, Santa Cruz de Tenerife, Spain. [5]These authors contributed equally: Yuritza Y. Escalante, Jenna N. Adams. ✉e-mail: njanssen@ull.edu.es

cortex confers geometric constraints to the functional dynamics of the brain, with a greater impact on activity than that of structural connectomes representing white matter[20]. To our knowledge, this framework of examining changes in the global geometry of the brain has not been applied to the study of age-related changes in brain structure. Critically, if the spatial geometry of the brain in fact impacts functional dynamics[20], these age-related changes to the spatial geometry of the brain may be one underlying mechanism associated with clinical impairment and age-related cognitive changes.

One compelling method to examine how the global geometry of the brain changes with age is using Euclidean distance, a measure of distance between the center of mass of points or regions. Euclidean distance is commonly used in quantifying shape differences in anthropological research[21]. Within neuroimaging, Euclidean distance has been used as a control variable in some graph analyses of structural or functional connections[22,23], but has rarely been the focus of analysis. However, quantifying changes in global brain shape and distances between regions using Euclidean distance could provide important insight into changes to the overall spatial geometry of the brain. Applying this method, we are able to determine how aging affects global patterns of expansion and compression of the brain, both across whole-brain gradients and between specific region pairs, which provides information about complex spatial relationships of brain structures.

## Table 1 | Demographics of the OASIS sample

| Demographic Variable | OASIS-3, N = 2039 |
| --- | --- |
| Age (years, M ± SD) | 71.25 ± 9.37 Range: 42.7 to 97.1 |
| Sex (female: n, %) | 1045 (56.67%) |
| Education* (n, %) | |
| High School | 1794 (97.3%) |
| Bachelor's Degree | 1,140 (61.8%) |
| Master's Degree | 616 (33.4%) |
| Doctorate Degree | 153 (8.3%) |
| Race/Ethnicity (n, %) | |
| White | 1,603 (86.9%) |
| Black | 223 (12.1%) |
| Hispanic/Latino | 11 (0.6%) |
| Other | 7 (0.4%) |
| CDR ( > 0) | 277 (22%) |
| Episodic Memory (z-score) | Range: -2.66 to 2.52 |
| Executive Function (z-score) | Range: -1.15 to 5.56 |
| Working Memory (z-score) | Range: -2.85 to 2.18 |

*Education was measured as the cumulative highest level of education, i.e if a person has a Bachelor's Degree, they are included in both the Bachelor's Degree and High School groups

In the current study, we seek to comprehensively determine how aging, from mid-life onwards, impacts the overall geometric shape of the brain as well as complex spatial relationships between regions. To test this, we examined 2039 structural MRIs of adults (aged 42–97 years) from the Open Access Series of Imaging Study (OASIS), as well as a replication set of 564 structural MRIs (aged 30–88 years) from the Cambridge Centre for Ageing and Neuroscience (Cam-CAN) study. We hypothesized that increasing age would be associated with a more severe pattern of geometric changes in the global shape of the brain and spatial relationships between regional homologues. Further, we hypothesized that global and regional changes to brain geometry would be associated with clinical impairment and cognitive performance, suggesting these changes in shape or spatial relationships between regions have functional consequences. Our results demonstrate spatially-specific gradients of expansion and compression within the aging brain that are exacerbated in older adults who are clinically impaired and perform worse on cognitive assessments, suggesting large-scale changes to the spatial geometry of the brain may contribute to the expression of cognitive decline.

## Results

### Aging is associated with global gradients of brain compression and expansion

To determine how aging affects overall brain shape, we analyzed 2039 structural T1 MRIs from 1059 unique participants from the Open Access Series of Imaging Studies 3 (OASIS-3) study[24]. Participants ranged from 42–97 years old (71.7 ± 9.2 years). Full participant demographic information is presented in Table 1. Structural T1-weighted scans were processed using the Human Connectome Project (HCP) minimal preprocessing pipeline (see Methods for full processing details) to prepare for analyses.

We first assessed how aging was associated with changes to the overall shape of the brain, measured along inferior-superior and anterior-posterior gradients. This was achieved by calculating global distances between equally spaced points on the outer cortical surface of the brain ($n = 400$), equally spaced along inferior to superior and anterior to posterior directions on the axial plane on the T1-weighted MRI (see Methods; Fig. 1A). The Euclidean distance between each cross-hemisphere pair of points was quantified.

To determine the effect of age on global distance across these inferior to superior and anterior to posterior gradients (Fig. 2), we constructed models predicting to distance, with sex, scan quality (Euler value), estimated total intracranial volume, and clinical status as covariates of no interest (see Methods and Table 2). We first assessed the effect of age when entered continuously. There was a significant effect of age on distances across inferior-superior and anterior-posterior gradients ($F(81,614299) = 19.46$, $p < 0.0001$, partial $\eta^2 = 0.003$, 95% CI [0.00, 1.00], Table 2). Older age was associated with

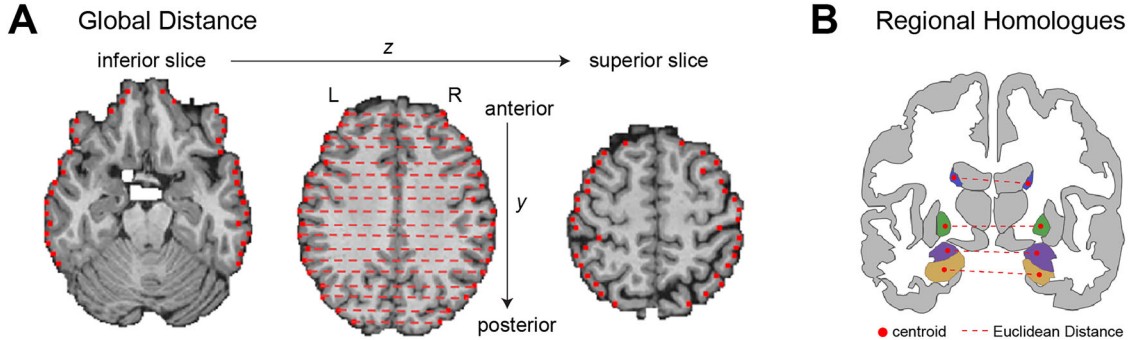

**Fig. 1 | Visualization of methods to compute global distances (A) and regional homologues (B). A** Gradients of whole-brain expansion and compression were quantified by placing equally spaced points on the outer cortical surface ($n = 400$) along inferior-superior ($z$) and anterior-posterior ($y$) directions. **B** Distance between regional homologues were determined by quantifying the Euclidean distance between the center of mass across left- and right- hemisphere homologous regions.

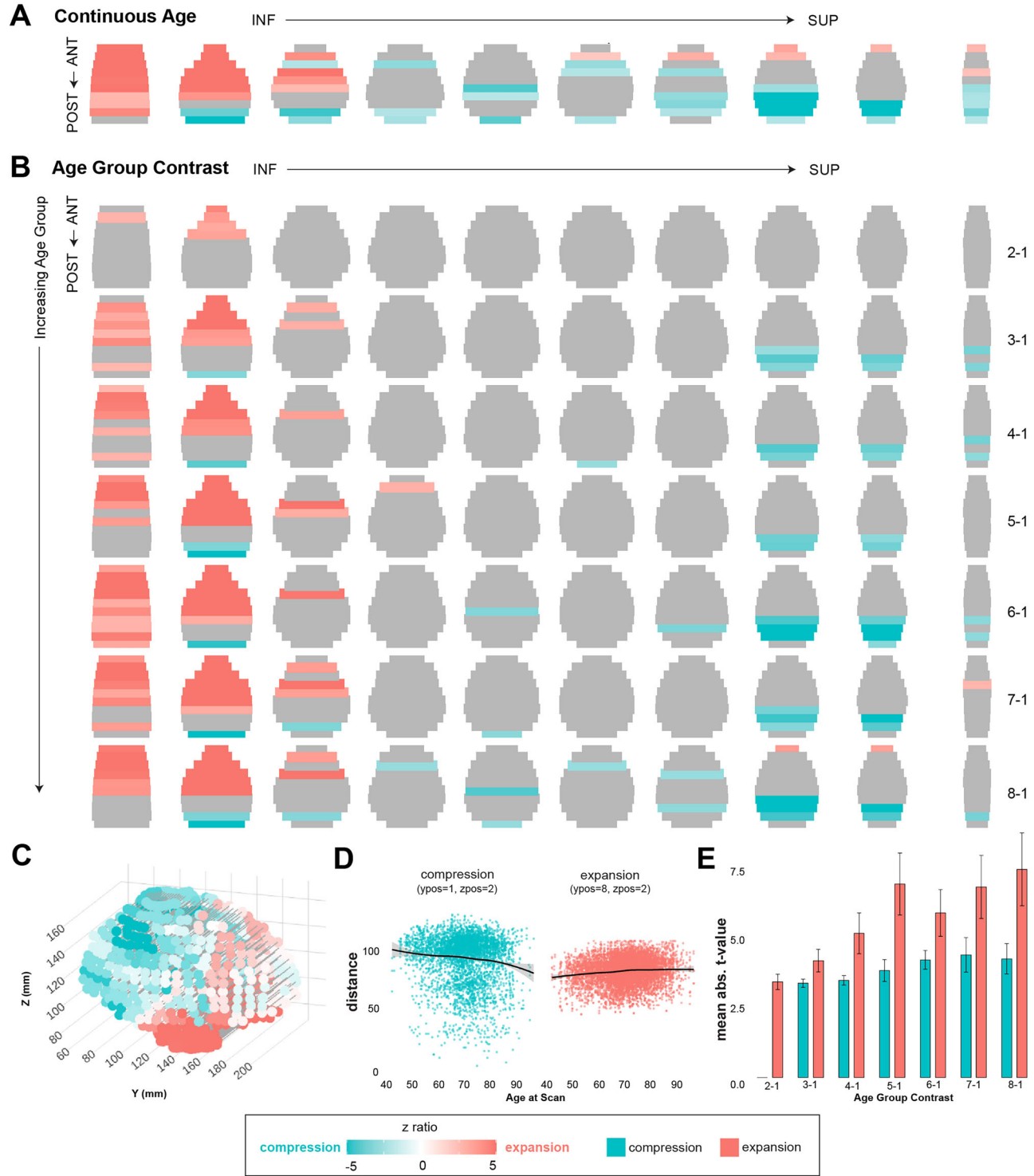

**Fig. 2 | Age-related changes to global distances between the outer edges of the brain.** Effects of age were assessed continuously (**A**) and by comparing discrete groups of increasing age (Groups 2–8) to a young control group (Group 1) to detect cross-sectional emergence patterns (**B**). Increasing age was associated with global expansion (red) in inferior-anterior regions and with compression (blue) in posterior-superior regions (**A**). These effects began with inferior-anterior expansion in the youngest age comparison (Group 2 compared to 1), and intensified with increasing age groups (**B**). Compression first emerged in superior-posterior regions (Group 3 compared to 1), and later affected the middle of the brain in older age groups. **C** 3D representation of the continuous age results shown in (**A**), and corresponds to the Supplemental Movie 1 file which shows a rotating view of these effects. **D** Expansion (red) and compression (blue) between the outer edges of the brain plotted as a function of continuous age. Regression smoother (LOESS) in grey with SE. **E** Expansion (red) and compression (blue) effect sizes compared across age groups demonstrates the earlier emergence and stronger effect of expansion compared to compression (n = 100 brain locations per bar). Data are represented as mean values +- SEM.

**Table 2 | Statistical results for the global distances analyses**

| Contrast | Sum of Squares | Mean Square | Num DF | Den DF | F | p |
|---|---|---|---|---|---|---|
| Continuous Age[a] | 192470.28 | 2376.18 | 81 | 614299.26 | 19.46 | 3.27e−275 |
| Age Group[a] | 239742.53 | 422.83 | 567 | 613704.93 | 3.46 | 1.21e−152 |
| Clinical Status[b] | 78491.16 | 969.03 | 81 | 614300.95 | 7.90 | 1.10e−87 |
| Episodic Memory[c] | 35315.60 | 436.00 | 81 | 290478.76 | 3.30 | 8.58e−22 |
| Executive Function[c] | 22034.04 | 272.032 | 81 | 290479.99 | 2.06 | 2.27e−08 |
| Working Memory[c] | 10308.21 | 127.26 | 81 | 290478.84 | 0.96 | 0.57 |

All models predicted to global distance along inferior to superior and anterior to posterior gradients, controlling for sex, estimated total intracranial volume, and scan quality. Statistics reflect the three-way interaction between the variable of interest, inferior-superior, and anterior-posterior gradients (Mixed effect regression). [a]Models additionally controlled for clinical status. [b]Model additionally controlled for age. [c]Model additionally controlled for age and clinical status.

the expansion within the inferior-anterior portion of the brain, and compression within the more posterior aspects of the brain in both inferior and superior locations (Fig. 2A, C and D; Table 2; Supplementary Movie 1). Interestingly, covariates of no interest such as total intracranial volume and sex did in fact demonstrate significant main effects on distance gradients (intracranial volume: $F_{(1,1978,)} = 774.84$, $p < 0.0001$, partial $\eta^2 = 0.28$, 95% CI [0.26, 1.00]; sex: $F_{(1,1977)} = 73.99$, $p < 0.0001$, partial $\eta^2 = 0.04$, 95% CI [0.02, 1.00]), however, further investigation of these factors were beyond the scope of the primary research question.

To further isolate when in the aging process these effects emerge, we split our sample into eight non-overlapping age bins with equivalent numbers of scans (see Supplementary Table 1). We then contrasted each increasing age bin (Groups 2−8) to the youngest age bin (Group 1) in similar models to the continuous age model (see Methods). There was also a significant effect of age group on distances across inferior-superior and anterior-posterior gradients ($F_{(567)} = 3.46$, $p < 0.0001$, partial $\eta^2 = 0.003$, 95% CI [0.00, 1.00], Fig. 2B and E; Table 2; Supplementary Movie 2). Follow-up age group contrasts demonstrated that inferior-anterior expansion emerged early in the aging process (Group 2−Group 1 contrast) and increased in spatial extent as age group increased. Compression first emerged in superior-posterior regions (Group 3−Group 1 contrast), and increased in spatial extent and also appeared within inferior-posterior regions as age group bin increased. Together, this pattern of results support an inferior-anterior gradient of global brain expansion and a superior-posterior gradient of global brain compression that is associated with the aging process.

### Clinical status and cognitive performance are associated with distinct patterns of global compression and expansion

We next aimed to determine if clinical status and cognitive performance were associated with similar patterns of expansion and compression, over and above the observed age-related effects (Fig. 3A). To test the effect of clinical status (Clinical Dementia Rating (CDR) of 0 compared to > 0), we constructed models predicting to distance, with clinical status as the predictor of interest, and including age, sex, scan quality, and estimated total intracranial volume as covariates of no interest. There was a significant effect of clinical status on distances across inferior-superior and anterior-posterior gradients ($F_{(81)} = 7.90$, $p < 0.0001$, partial $\eta^2 = 0.001$, 95% CI [0.00, 1.00], Table 2). Clinical impairment (CDR > 0) was associated with patterns of compression in inferior-anterior regions and expansion in posterior regions across inferior, middle, and superior regions while controlling for effects of age (Fig. 3A).

We then assessed how three different domains of cognitive performance (episodic memory, executive function, and working memory; see Methods) were associated with global distance change, controlling for effects of both age and clinical status in models. There was a significant effect of episodic memory ($F_{(81)} = 3.30$, $p < 0.0001$, partial $\eta^2 = 0.0009$, 95% CI [0.00, 1.00]) and executive function ($F_{(81)} = 2.06$, $p < 0.0001$, partial $\eta^2 = 0.0005$, 95% CI [0.00, 1.00]) on

distances across inferior-superior and anterior-posterior gradients. Episodic memory performance was only associated with change within inferior regions, with worse performance associated with expansion anteriorly and compression posteriorly (Fig. 3B). In contrast, worse executive function performance was associated with expansion in inferior-anterior regions and compression within inferior and middle regions (Fig. 3C). Finally, working memory did not exhibit a significant effect on whole brain gradients ($F_{(81)} = 0.96$, $p = 0.57$; Table 2; Fig. 3D shows effects but did not reach corrected significance).

### Spatial relationships between regional homologues change with age

To further determine how these global patterns of expansion and compression arose, we next investigated how spatial relationships between regional homologues, or equivalent grey matter regions across opposite hemispheres, change with age. To do this, we identified the center of mass of each of 78 unilateral regions of interest (39 homologous pairs) spanning cortical and subcortical Freesurfer regions, and calculated the Euclidean distance between these homologous pairs (see Fig. 1B; Methods). Then, we constructed models predicting to distance, with age as the predictor of interest, and including clinical status, sex, scan quality, and estimated total intracranial volume, and volume of each ROI as covariates of no interest. Importantly, regional volume was included as a covariate to ensure the results did not simply reflect regional atrophy seen with aging.

Increasing age was overwhelmingly associated with expansion between regional homologues ($F_{(38)} = 102.44$, $p < 0.0001$, partial $\eta^2 = 0.05$, 95% CI [0.05, 1.00]; Table 3). The strongest age-related expansion effects occurred between the homologues of subcortical regions, specifically the caudate nucleus, thalamus, amygdala, pallidum, and putamen (Fig. 4A; Supplementary Fig. 1). Expansion also strongly occurred between homologues of age-vulnerable medial and lateral temporal regions such as the entorhinal cortex, hippocampus, parahippocampal cortex, and superior, middle, and inferior temporal cortex. The only region to demonstrate significant compression between homologues was the superior parietal cortex. Posterior-superior midline regions, such as precuneus and retrosplenial (isthmus cingulate) cortex, did not demonstrate significant age-related changes. To determine whether ventricular enlargement explained the preferential expansion, particularly of midline and subcortical regions, we re-ran this model with additionally including ventricular volume. Controlling for ventricular volume, results were largely consistent with initial models, with subcortical regions still demonstrating very significant effects of expansion, but slightly reduced effect sizes compared to initial results (see Supplementary Fig. 2).

To further stage the progression of age-related regional homologue distance changes, we again constructed a model examining increasing age groups (Groups 2−8 contrasted against Group 1). The effect of age group was also significantly associated with regional homologue distances ($F_{(38)} = 15.87$, $p < 0.0001$), partial $\eta^2 = 0.05$, 95% CI [0.05, 1.00]; Table 3. Follow-up age group contrasts (Fig. 4B) indicated that expansion

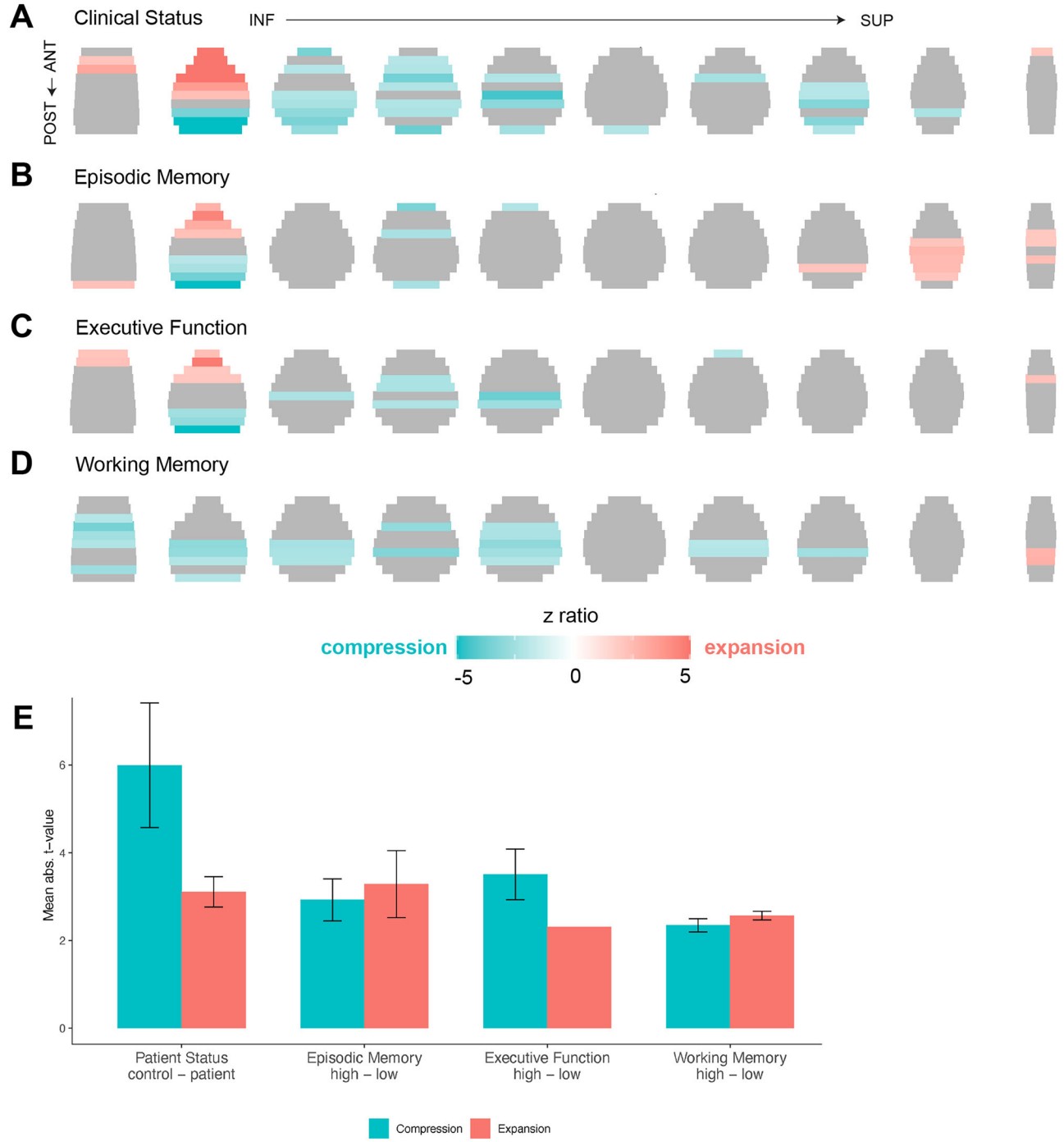

**Fig. 3 | Clinical status and cognitive-related changes to global distances between the outer edges of the brain.** Clinical impairment was defined as CDR > 0. Episodic memory, executive function, and working memory were measured with neuropsychological assessments of Logical Memory, Trail Making Test B, and the Digit Span test, respectively. All models for clinical impairment and cognitive performance controlled for effects of age. Whole brain gradients of expansion (red) and compression (blue) were associated with clinical impairment (**A**), episodic memory (**B**), and executive function (**C**). Effects related to working memory did not reach significance, but subthreshold results are shown in (**D**). **E** Comparison of effect sizes of expansion (red) and compression (blue) across the different domains (*n* = 100 brain locations per bar). Data are represented as mean values +- SEM.

between regional homologues first occurred in subcortical and temporal lobe regions, largely spanning the inferior portion of the brain (Group 2-1 contrast). With increasing age (Group 5-1 contrast), expansion then intensified within these regions, and additionally included occipital and frontal regions. Finally, in the oldest age group (Group 8-1 contrast), expansion intensified in occipital and frontal regions, and compression emerged in the superior parietal cortex. Full trajectories of expansion and compression by age group are shown in Supplementary Fig. 3. This

cross-sectional staging approach demonstrates a complex, regionally and temporally specific pattern to to changes in spatial relationships between regional homologues.

**Spatial relationships between regional homologues change with clinical status and are related to cognitive performance**

We again sought to determine if observed changes in spatial relationships between regional homologues were further related to

**Table 3 | Statistical results for the regional homologues analyses**

| Contrast | Sum of Squares | Mean Square | Num DF | Den DF | F | p |
|---|---|---|---|---|---|---|
| Continuous Age[a] | 19789.89 | 520.79 | 38 | 74370.61 | 102.44 | 2.22e−308 |
| Age Group[a] | 21438.56 | 80.60 | 266 | 74063.77 | 15.87 | 2.22e−308 |
| Clinical Status[b] | 6938.00 | 182.58 | 38 | 74370.83 | 34.73 | 5.37e−250 |
| Episodic Memory[c] | 2167.06 | 57.03 | 38 | 35187.49 | 10.74 | 4.07e−63 |
| Executive Function[c] | 2376.27 | 62.53 | 38 | 35187.91 | 11.79 | 5.82e−71 |
| Working Memory[c] | 1327.60 | 34.94 | 38 | 35186.75 | 6.55 | 1.15e−32 |

All models predicted to the Euclidean distance between regional homologues, controlling for sex, estimated total intracranial volume, and scan quality. Statistics reflect the interaction between the variable of interest and region (Mixed effect regression). [a]Models additionally controlled for clinical status. [b]Model additionally controlled for age. [c]Model additionally controlled for age and clinical status.

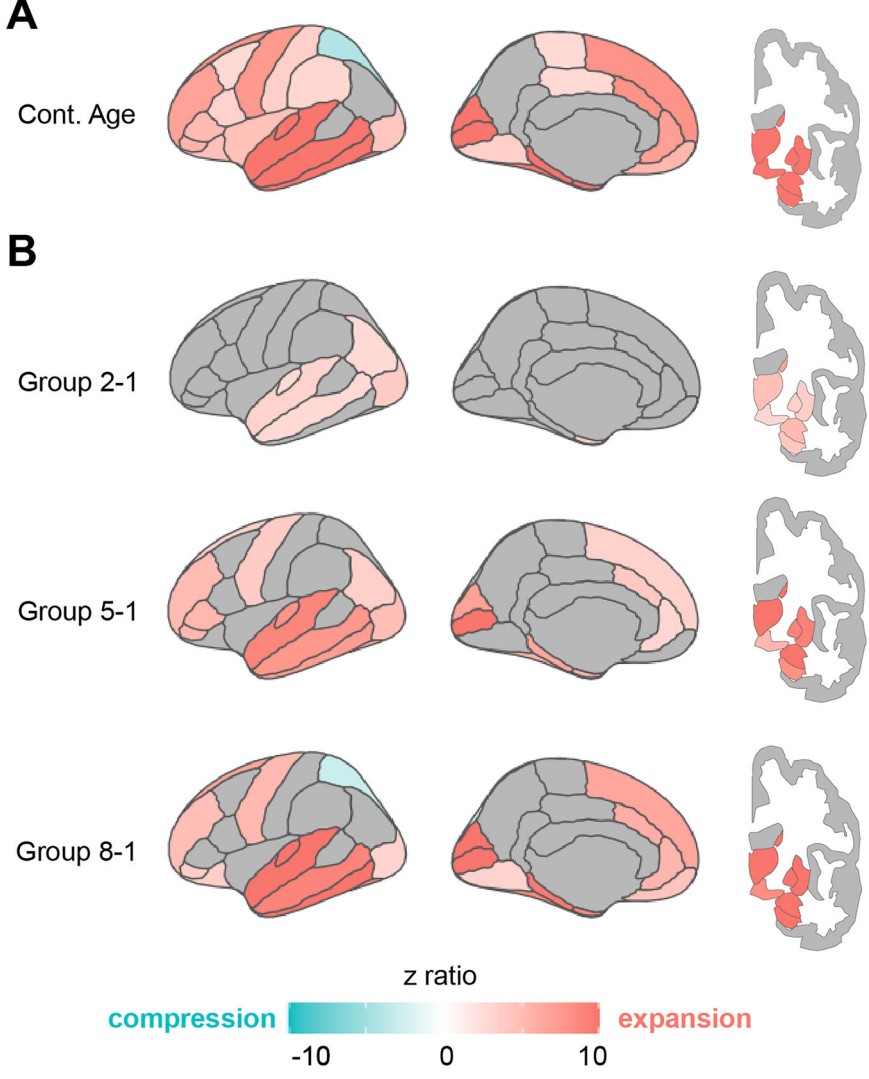

**Fig. 4 | Age-related changes in spatial relationships between regional homologues.** The Euclidean distance between the center of mass of regional homologues was quantified. **A** Association between regional distances and continuous age demonstrates that expansion between homologues primarily occurs with older age, strongest in temporal and frontal regions. **B** Comparisons of increasing age groups (Groups 2 through 8) to a younger control group (Group 1) to cross-sectionally model temporal effects. Expansion (red) first occurs in subcortical, medial, and lateral temporal regions (Group 2-1 contrast). Expansion then becomes stronger in these regions, and also includes frontal and occipital regions (Group 5-1 contrast). Finally, expansion continues to become stronger, and lateral parietal compression (blue) occurs (Group 8-1 contrast).

clinical impairment and cognitive performance. First, to assess effects of clinical status on regional distances, we constructed a model predicting distance, with clinical status as the predictor of interest, and including age, sex, scan quality, estimated total intracranial volume, and regional volumes as covariates of no interest. The effect of clinical status on regional distances was significant ($F(38) = 34.73$, $p < 0.0001$,

partial $\eta^2 = 0.02$, 95% CI [0.02, 1.00]; Table 3). Consistent with the age-related effects, clinical impairment was associated with expansion of subcortical and medial and lateral temporal lobe regions (Fig. 5A). However, clinical impairment was uniquely associated with compression of medial parietal and occipital regions, regions which failed to show an age-effect or exhibited expansion. This medial, superior-

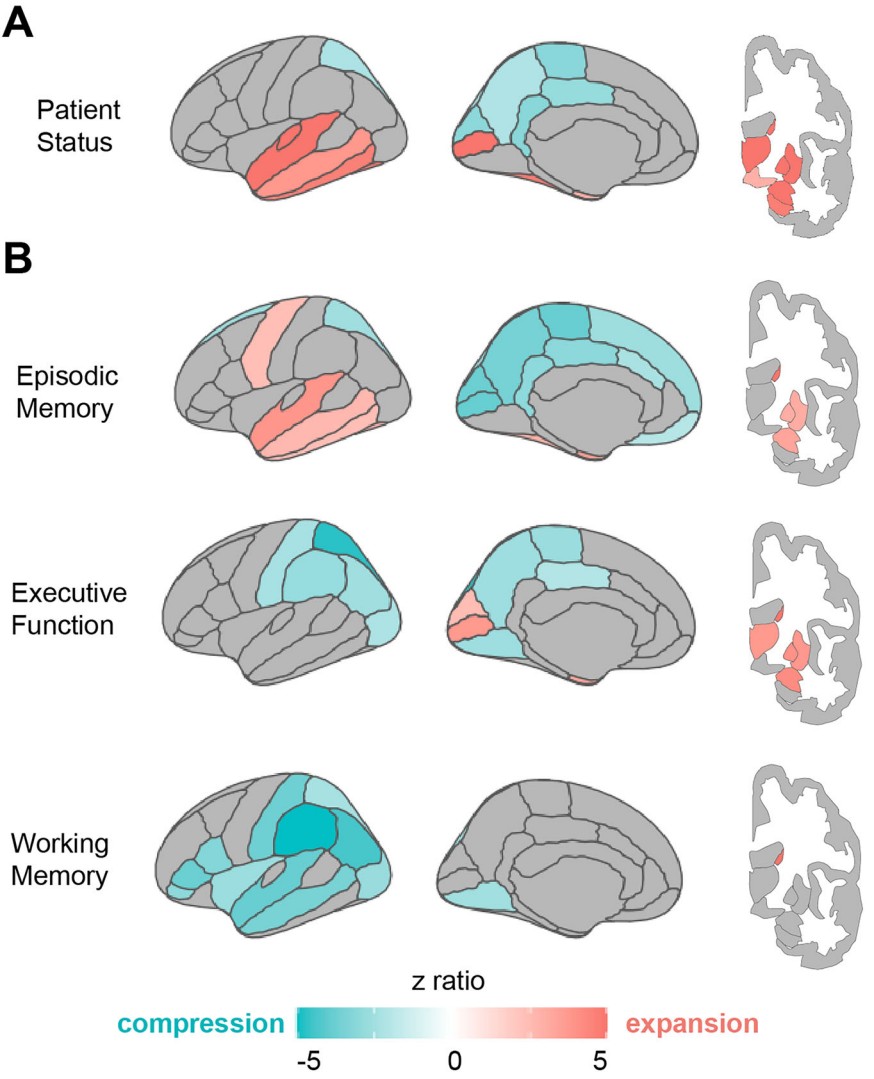

**Fig. 5 | Clinical and cognitive-related changes in distances between regional homologues.** All models for clinical impairment and cognitive performance controlled for effects of age. **A** Clinical impairment (patient status, defined as Clinical Dementia Rating scale = 0 for Control and Clinical Dementia Rating Scale > 0 for patients) was associated with expansion (red) in subcortical, medial, and lateral temporal regions, and with compression (blue) in medial parietal regions.

**B** Distinct patterns emerged across the different cognitive domains. Worse episodic memory performance was associated with subcortical, medial, and lateral temporal expansion, with compression across medial parietal and medial frontal regions. In contrast, worse executive function was only associated with compression across medial and lateral parietal and occipital regions, while working memory was associated with compression in lateral parietal and temporal regions.

posterior regional compression most strongly occurred within pericalcarine cortex, retrosplenial, paracentral, and cuneus cortex (Supplementary Fig. 1).

Finally, we investigated whether cognitive performance, over and above age and clinical status, was also associated with changes in spatial relationships between regional homologues. There was a significant effect of all three cognitive domains on regional distance (episodic memory: $F(38) = 10.74$, $p < 0.0001$, partial $\eta^2 = 0.01$, 95% CI [0.01, 1.00]; executive function: $F(38) = 11.79$, $p < 0.0001$, partial $\eta^2 = 0.01$, 95% CI [0.01, 1.00]; working memory: $F(38) = 6.55$, $p < 0.0001$, partial $\eta^2 = 0.007$, 95% CI [0.00, 1.00]; Table 3). Strikingly, the observed pattern of expansion and compression of regional homologues varied by cognitive domain, and mapped onto regions known to contribute to the underlying cognitive process (Fig. 5B–D; Supplementary Fig. 1). For example, worse episodic memory was uniquely associated with expansion of temporal lobe regions, as well as compression of medial parietal regions. These patterns largely resembled the effects of clinical impairment, even with clinical status being controlled for in the model (Fig. 5B). In contrast, worse executive function was defined by both

medial and lateral parietal compression, and expansion largely limited to subcortical and medial occipital regions (Fig. 5B). Finally, working memory was associated with a widespread pattern of lateral compression, especially within lateral parietal, temporal, and frontal regions, with limited evidence of subcortical expansion, other than caudate nucleus (Fig. 5B). These regionally specific patterns suggest various cognitive processes are differently affected by expansion and compression of regional homologues.

## Sensitivity analyses
To ensure reliability of our results, we performed a number of sensitivity analyses. First, to determine if the number of sampled points (20 × 20 locations, 400 points) placed for global distance analysis influenced observed results, we repeated analyses with a smaller (15 × 15 locations, 225 points) and greater (30 × 30 locations, 900 points) number of points along the outer edge of the brain. The association between global shape change and age remained significant regardless of the number of points (225 points: $F(81,324968) = 10.03$, $p < 0.0001$, partial $\eta^2 = 0.003$, 95% CI [0.00, 1.00]; 900 points: $F(81,1455126) = 37.72$, $p < 0.0001$, partial

$\eta^2 = 0.002$, 95% CI [0.00, 1.00]), and spatial gradients of expansion and compression were generally consistent with the original results, especially when comparing 900 sampled points to the original parameter of 400 sampled points (Supplementary Fig. 4A; see Supplementary Table 3 for quantification of spatial correlation across parameters and analyses). Next, we determined if downsampling the number of points during statistical analyses (20 points to 10 points to facilitate analytic computations) influenced our results. With no downsampling in the statistical models, we observed convergent results with our original analyses ($F(361,1454519) = 18.52$, $p < 0.0001$, partial $\eta^2 = 0.005$, 95% CI [0.00, 1.00]; Supplementary Fig. 4B).

To test internal validity of our analyses, we performed bootstrap analysis with replacement for all primary analyses (global distance and homologues for age, clinical status, and cognitive domains). Bootstrapped results demonstrated no significant bias in overestimation of the observed results, as the observed effect size fell within the 95% confidence interval (see Supplementary Fig. 5).

To ensure our global expansion and compression results were not driven by the influence of gyrus to sulcus or sulcus to sulcus point pairs, we restricted our analyses to confirmed gyrus to gyrus pairs (see Methods) according to the Destrieux Atlas. Gyrus to gyrus pairs accounted for 88% of observed distance measurements, while a minority of measurements was represented by gyrus to sulcus (11%) or sulcus to sulcus (1%) point pairs. Re-analyzing our data using distance measurements restricted to the gyrus to gyrus pairs demonstrated very consistent results for the effects of continuous age ($F = 2.34$, $p < 0.0001$, partial $\eta^2 = 0.001$, 95% CI [0.00, 1.00]; spatial correlation with original results = 0.965), age group ($F = 1.14$, $p < 0.02$, partial $\eta^2 = 0.004$, 95% CI [0.00, 1.00]; spatial correlation with original results = 0.968), and fluid intelligence ($F = 1.12$, $p < 0.05$, partial $\eta^2 = 0.0006$, 95% CI [0.00, 1.00]; spatial correlation with original results = 0.961).

One potential limitation of using Euclidean distance to measure the distance between regional homologues is that asymmetry in location may result in a non-horizontal line, which may bias results. To test this, we calculated the total deviation of the line between each pair (see Methods), observing the overall total deviation from a straight line was relatively minor with few outliers (Supplementary Fig. 6A, B). Replicating our main results while including the degree of deviation for each measurement resulted in a non-significant interaction of deviation by region by age ($p = 0.2$), and restricting analyses to homologue pairs with deviation scores of < 10 did not impact the crucial region by age interaction (restricted data $F = 101.67$, $p < 0.0001$, partial $\eta^2 = 0.05$, 95% CI [0.05, 1.00]; Supplementary Fig. 6C; spatial correlation with original results, $r = 0.999$). Together, these control analyses suggest that the degree to which the homologue distance deviations from a straight line does not impact our results.

Finally, we attempted to further validate our cross-sectional results by exploiting the longitudinal component of the OASIS3 dataset. Post-hoc construction of a longitudinal dataset by maximizing the number of years between baseline and follow-up scans yielded 499 individuals whose mean age was 68.5 years old (SD = 9.6 years) at baseline and 73.5 years old (SD = 8.5 years) at follow-up. We performed an analysis modelling longitudinal change in global distance and regional homologue expansion and compression to further validate our cross-sectional results. The results were in line with our earlier observations for both global distance ($F(81,309996) = 2.68$, $p < 0.0001$, partial $\eta^2 = 0.0007$, 95% CI [0.00, 1.00]) and homologues ($F(38,37260) = 4.03$, $p < 0.0001$, partial $\eta^2 = 0.004$, 95% CI [0.00, 1.00]; see Supplementary Fig. 7). Future longitudinal work with increased time between timepoints should further examine this issue.

### Global and regional distance effects replicate within an independent dataset

To further verify the robustness and consistency of our results, we replicated key analyses within the Cambridge Centre for Ageing and Neuroscience (Cam-CAN) dataset, an open-access dataset with structural MRI and cognitive measures within healthy older adults[25,26]. The included Cam-CAN sample ($N = 564$) ranged from 30–88 years, with a mean age of 54.01. We restricted scans within the Cam-CAN dataset to range from at least 30 years of age to ensure results did not reflect early-adulthood development of brain structures, and to more closely parallel the age range found within OASIS while keeping a robust sample size. However, the Cam-CAN sample still skewed younger than the range of the OASIS sample, which was adults of at least 42 years of age with a mean of 71.25 years (see Supplementary Table 3 for demographics of included Cam-CAN participants).

We first conducted parallel analyses as done within the OASIS dataset to assess how global distance gradients were related to continuous age and age group contrasts. The effect of continuous age ($F(81) = 3.04$, $p < 0.0001$, partial $\eta^2 = 0.001$, 95% CI [0.00, 1.00]; Supplementary Table 4) and age group ($F(567) = 1.17$, $p = 0.003$, partial $\eta^2 = 0.003$, 95% CI [0.00, 1.00]; Supplementary Table 4) were highly significant. We found strikingly similar patterns of increasing age related to expansion of inferior-anterior aspects and compression related to superior-posterior aspects of the brain (Supplementary Fig. 8A). The age group contrasts demonstrated a similar pattern of inferior-anterior expansion within the youngest groups (Groups 2-1 & 3-1 contrasts), and then widespread superior-posterior compression in contrasts representing older age groups (Supplementary Fig. 8B). These results were very similar to that within the OASIS sample (Fig. 2).

We next investigated distances between regional homologues in the Cam-CAN dataset. The effect of continuous age ($F(41) = 24.22$, $p < 0.0001$, partial $\eta^2 = 0.04$, 95% CI [0.04, 1.00]; Supplementary Table 4) and age group contrasts ($F(41) = 4.56$, $p < 0.0001$, partial $\eta^2 = 0.05$, 95% CI [0.05, 1.00]; Supplementary Table 4) were again significant. The regional homologue results also closely resembled the OASIS dataset, with increasing age primarily associated with expansion, most strongly in subcortical and temporal regions (Supplementary Fig. 9A). Increasing age groups demonstrated expanded spatial extent and intensity of expansion across the brain, generally consistent with the OASIS findings (Fig. 4).

Because Cam-CAN was limited to cognitively healthy older adults, we were not able to replicate results regarding clinical status. However, we did test relationships between global distances and regional homologues with cognition. The Cam-CAN dataset had different cognitive measures than OASIS, without equivalent proxies for episodic memory, executive function, and working memory. Therefore, we focused on a measure of fluid intelligence to assess general cognition. Fluid intelligence was significantly related to both global distance ($F(81) = 1.71$, $p < 0.0001$, partial $\eta^2 = 0.0008$, 95% CI [0.00, 1.00]; Supplementary Table 4) and distance between regional homologues ($F(41) = 13.20$, $p < 0.0001$, partial $\eta^2 = 0.02$, 95% CI [0.02, 1.00]; Supplementary Table 4). Associations between fluid intelligence with both global distances (Supplementary Fig. 8C) and regional homologues (Supplementary Fig. 9C) showed similar patterns to that of increasing age, consistent with fluid intelligence being a more generalized measure of cognitive function rather than relying on any specific group of regions.

Overall, our results are closely replicated in an independent dataset with variation in age, demographics, and cognitive measurements, supporting the robustness of our findings.

## Discussion

The current study examined how aging is associated with widespread and complex patterns of change to the spatial geometry of the brain. We demonstrate that the overall shape of the brain undergoes gradients of change with increasing age, with expansion along the inferior-anterior aspect and compression along the superior-posterior aspect. Complex spatial relationships between regional cross-hemisphere homologues primarily demonstrate expansion in

subcortical and temporal lobe homologues, consistent with the inferior-anterior gradient across the whole brain. These age-related patterns are exacerbated in older adults who are cognitively impaired, and are differentially related to key domains of cognition in a regionally-specific manner mapping to regions underlying these cognitive processes. These patterns emerged in two independent datasets (OASIS, Cam-CAN) with varying age ranges and replicated using a within-subject longitudinal design and across various methodological parameters, demonstrating the robustness of these findings. Together, our results demonstrate that the brain undergoes spatially complex gradients of geometric structural change during aging, and suggest that the age-related disruption of normal geometry of the brain may contribute to cognitive impairment.

Our findings provide insights that build upon existing literature examining structural changes in the brain during aging by applying a unique approach to quantify gradients of expansion and compression across global brain shape and spatial distances between cross-hemisphere homologous pairs. Previous work characterizing gross anatomical changes of the aging brain have largely focused on volumetric or thickness changes to regions in isolation[6,13,27–29]. Although these studies highlight reduced volume in age-vulnerable regions such as the hippocampus, subcortical structures, and prefrontal cortex, this standard methodology does not characterize the relative change in position between these structures. In contrast, we extend volumetric work by demonstrating that these same structures are prone to relative spatial changes, with expansion in inferior-anterior and compression in superior-posterior aspects of the brain. Expansion predominantly targeted inferior-anterior regions such as the medial temporal lobe and subcortical regions. In contrast, compression was observed in superior-posterior aspects of the brain. Our findings have a number of implications for the study of age-related structural changes. First, they reinforce that considering the global shape and spatial geometry of the brain provides complementary information to the study of regional atrophy. Our results also may provide a basis for previously observed age-related changes in compression versus expansion in skull shape[30], which are thought to result from mechanical forces of the brain[31]. Finally, we demonstrate these global patterns of expansion and compression have cognitive implications even when controlling for age and regional atrophy patterns, suggesting the spatial geometry of the brain may strongly constrain brain function[18–20].

One noteworthy finding that emerged from the gradient approach we employed was of expansion within inferior-anterior portions of the brain, which correspond to inferior/anterior frontal and temporal regions. In our understanding of the previous literature, there is little evidence suggesting that brain volume in these regions *increase* with age; in fact, to the contrary, these regions are some of the most vulnerable to neurodegeneration in aging and neurodegenerative disease[2,11,14,15,32–36]. Because our approach measured changes to spatial relationships between distinct regions, it may have increased sensitivity to detect previously undescribed mechanisms of structural changes. While each individual region may be decreasing in volume with age, reflecting atrophy, the overall shape of the brain may "slouch" along the inferior-anterior aspect, which expresses as expansion of distances between the outer edges and regional structures. Consistent with this hypothesis, our results were robust to total brain volume, ventricle size, and the inclusion of regional volumes within homologue models. Whether this "slouching" phenomena is a result of the breakdown of anatomical connections, microscopic support structures, or even reflects the effects of gravity across the lifespan should be further investigated in subsequent studies.

Our age-group analysis provided a cross-sectional model of the temporal emergence of the global and regional expansion and compression patterns within the aging brain. Expansion was first observed in subcortical, medial temporal lobe, and lateral temporal lobe regions.

In older age groups, expansion then emerged in frontal brain regions. The brain region that showed the strongest effect with age was the caudate nucleus, which has previously been shown to atrophy with age[12,13,37], though other subcortical and medial temporal regions also showed strong effects. We tested whether ventricular enlargement may explain the preferential expansion of the subcortical regions, which are situated along either side of the lateral ventricles. However, even when controlling for ventricle size, the caudate nucleus and other midline subcortical structures still demonstrated highly significant effects (Supplementary Fig. 2), suggesting that these findings are not solely explained by ventricular enlargement.

While these patterns of compression and expansion seem to be characteristic of the "normal" aging process, they were exacerbated within clinically impaired populations and may therefore contribute to cognitive decline. When comparing older adults with clinical impairment to those with normal cognition, controlling for age, similar though enhanced patterns of whole brain and regional homologue expansion emerged. We found consistent evidence of expansion of inferior-anterior aspects of the overall brain shape, paired with increase in distance between homologues of subcortical and medial temporal regions. Interestingly, as mentioned above, medial temporal, subcortical, and medial parietal regions are especially prone to atrophy in neurodegenerative disease[36]. In contrast, we found compression of posterior aspects of the overall brain shape, paired with a decrease in distance between medial parietal homologues, such as the superior parietal, precuneus, and cuneus regions. This compression of regional homologues was unique to clinical impairment, and not found to be associated with increasing age. The imbalance between inferior-anterior expansion and parietal compression may be a phenotype of clinical impairment. For example, creation of reference curves of global compression and expansion patterns associated with "normal" aging, such as done with Brain Age Gap Estimation models[36,38], could be created from larger datasets and used to identify individual participant's patterns of compression or expansion that may be related to known neurodegenerative disorders or clinical impairment.

Furthermore, we examined how various cognitive domains, specifically episodic memory, executive function, and working memory, are related to these patterns while also controlling for effects of age and overall clinical status. Worse episodic memory and executive function performance were further related to similar whole brain patterns as shown to be related to the aging process and clinical impairment. However, we found a striking pattern of regional specificity in how spatial relationships between homologues relates to cognitive performance. Episodic memory was associated with a pattern highly resembling overall clinical impairment, with inferior-anterior expansion targeting subcortical and temporal regions known to support memory, and compression in posterior and parietal regions. In contrast, executive function was primarily associated with compression in medial parietal regions, while working memory was associated with lateral parietal, frontal, and temporal regions, reflecting the distributed nature of working memory[39,40]. In our replication dataset, we investigated associations with fluid intelligence, a general measure of cognitive function. The pattern associated with fluid intelligence was very similar to the overall pattern of aging, and did not demonstrate the regional specificity of the other cognitive domains, consistent with the generalized nature of fluid intelligence[41]. Overall, the distinct expansion and compression patterns related to different cognitive domains suggest that specificity in the location of geometric changes may be expressed as deficits in cognitive domains that rely on those particular regions to function.

One limitation of interpretation of results within this study is that a number of mechanistic contributors could have led to the observed patterns of whole brain and regional expansion and compression. Although our analyses ruled out that ventricular expansion, total intracranial volume, or regional atrophy accounted for the observed

expansion effects, other biological changes may impact geometric brain shape changes, such as unihemispheric brain changes or specific changes to white or grey matter. Therefore, future directions include the examination of how features such as white matter structural connections between regions[42,43], regional microstructural properties such as demyelination[44], vascular integrity[45,46], synaptic loss[47,48], and other molecular contributors that affect structural scaffolding in the brain may contribute to our findings. Further, an investigation on how age-related geometric differences in shape affect functional dynamics[20,49], such as neural oscillations and functional connectivity, may provide more insight on how these processes lead to age-related cognitive decline in older adults.

There are a number of methodological limitations that are important to highlight and address in future work. Factors such as the individual variability in head size and shape or gyrification patterns may have influenced our results, since analyses were performed in native space as necessary with this method. However, note that we thoroughly demonstrated that our results were robust to methodological parameters, such as the number of points placed for global distance calculation and downsampling for statistical analyses. In addition, our results replicated in a post-hoc longitudinal analysis as well as within an independent dataset (Cam-CAN) and demonstrated no bias from internal validation measures. A second potential concern relates to susceptibility artifacts on T1-weighted MRI that may primarily affect inferior-anterior aspects of the brain due to the close proximity to the skull base. This may have influenced our results, although linking such artifacts to changes in age, clinical status and cognition is not obvious and would not predict the observed expansion effects in brain regions relatively far removed from the skull base (i.e., caudate, thalamus). Nevertheless, addressing these issues in future studies will further solidify the results obtained here.

Future extensions of this work will provide more insight into specific degenerative conditions associated with aging, as well as a more precise understanding of how patterns of expansion and compression of the spatial geometry of the brain express as cognitive changes. We classified all older adults demonstrating impairment on the Clinical Dementia Rating into one group, though we recognize this group may represent a heterogeneous population. Future studies should also further investigate how different types of neurodegenerative disease (e.g. Alzheimer's disease versus frontotemporal dementia) specifically map onto these patterns of expansion and compression, as well as the impact of age- and disease-related pathology (e.g., amyloid-beta, tau, TDP-43)[50,51]. In particular, tau pathology, seen in both aging and degenerative diseases such as Alzheimer's disease[50], is closely related to regional patterns of atrophy in medial and lateral temporal lobe regions[52] that were found to undergo cross-hemisphere expansion with older age and memory impairment in the current study. Further, a more fine-grained investigation of different cognitive domains, or even the use of robust composite scores of various domains, would be an important next step to determine how these patterns may differentially express as various changes in cognition. Finally, we were not powered to conduct a robust longitudinal assessment of within-subject change in expansion and compression patterns alongside concurrent change in clinical status and/or cognition. Future studies should determine the temporal trajectory of structural versus cognitive changes to better assess the predictive value of imbalances between expansion and compression.

The current study provides insights of how the complex spatial geometry of the brain changes during aging. We demonstrate that aging is associated with distinct patterns of expansion and compression across different brain gradients, particularly affecting fronto-temporal regions. More importantly, we show that these morphological changes are strongly linked to cognitive dysfunction, suggesting that changes in the spatial geometry of the aging brain may have consequences for the efficiency of communication across brain regions that could underlie cognitive decline with aging. Our findings may be valuable in a clinical setting, as structural T1-weighted MRIs are standard and fairly straightforward to acquire, and often collected for diagnosis. However, more research is needed to solidify that variation in these patterns of geometric changes are indicative of cognitive impairment, and to test the predictive nature of this effect. Therefore, characterizing age-related geometric changes to the shape of the brain provides a unique approach to understanding cognitive deficits and a tool that could potentially be used in clinical practice.

## Methods
### Datasets
Two primary datasets were used for the analyses reported here. First, the Open Access Series of Imaging Studies (OASIS-3) dataset[24] is an extensive multifaceted dataset that brings together a large number of different clinical and imaging variables. Although the dataset has a longitudinal component and generally consists of multiple data acquisitions collected at consecutive timepoints, the current study considers the dataset in a cross-sectional manner. At the time when the data were downloaded (09-2020), 2168 MR imaging sessions from a total set of 1098 unique participants were available. As a consequence of preprocessing, actual analyses were carried out on a subset of 2039 MR sessions with 1059 unique participants. Of this final set, 588 were female and 471 were male. The mean age was 71.7 years and the standard deviation was 9.2 years. Given that this dataset was used in the primary analyses we will refer to this dataset as the test dataset.

The second dataset was the Cambridge Centre for Ageing and Neuroscience (Cam-CAN) dataset[25,26]. This is a smaller dataset that is purely cross-sectional. This dataset had 653 unique participants. As before, preprocessing requirements reduced this dataset to 642 participants. Within this final set, there were 325 females and 317 males and the average age was 54.1 years and the standard deviation was 18.4 years. Given that this dataset was used for validation purposes we will refer to this dataset as the replication dataset.

### MRI data and preprocessing
As per OASIS reference publications[24], all MRI data was collected through the Knight Alzheimer Research Imaging Program at Washington University in St. Louis, MO, USA. Some of the MRI data was collected on a Siemens Vision 1.5 T, while the majority of the scans came from two different versions of a Siemens TIM Trio 3 T (Siemens Medical Solutions USA, Inc). Participants were lying in the scanner in a supine position, head motion was minimized by inserting foam pads between the participant's head and antenna coil, and for some participants a vitamin E capsule was placed over the left temple to mark lateralization. A 16 channel head coil was used in all scans. Although a variety of different structural and functional imaging protocols are included in the OASIS dataset such as FLAIR, DTI and ASL, here we focused on the T1w and T2w scans. One important aspect of the OASIS dataset is that on a large number of occasions, multiple T1w or T2w images were collected for a given participant within a given MR session. Specifically, from the total number of 1957 MR sessions that we had available (see above), for 828 sessions there was a single T1w image available, while for the remaining 1129 sessions there was more than a single T1w image available. In detail, for 901 sessions there were two T1w scans, for 194 sessions there were three T1w scans, for 29 sessions there were four T1w scans, and for 5 sessions there were five T1w scans collected for a given participant within the same scanning session. As we describe below, one aspect of the HCP pipeline is that in such cases the individual T1w images for a given participant are co-registered and averaged into a single image to improve SNR. The T1w images were acquired using a 3DMPRAGE protocol TI/TR/TE: 1000/2400/3.08 ms, flip angle = 8°, resulting in 1 mm isotropic voxels. In addition, T2w images were acquired using a 3D SPC protocol: TR/TE: 3200/0.455 ms, flip angle = 120°, resulting in 1 mm isotropic voxels.

Preprocessing of the T1w and T2w images was done using the Human Connectome Project (HCP) minimal preprocessing pipeline (v4.1.3)[53]. The structural pre-processing pipeline of the HCP consists of a set of scripts that rely on programmes like FSL[54] and the Connectome WorkBench (Marcus et al., 2011). For the preprocessing of the T1w images we used the pre-freesurfer, freesurfer and post-freesurfer batch scripts with options that were applicable to our situation (i.e., no readout distortion correction, 1 mm MNI HCP structural templates, GRE fieldmap correction). The exact pre-processing steps executed during the pre-freesurfer script were (1) averaging repeated T1w and T2w acquisitions if such exist; (2) Perform alignment to ACPC axes in native structural space and undistort images with GRE fieldmaps; (3) Perform an initial robust brain extraction using FNIRT; 4. Align the T1w and T2w images; (5) Perform bias field correction. For the present purposes this preprocessing produced an image of the cortical ribbon

neuropsychological data (total intracranial volume performance on cognitive tests, etc). We ensured that all neuropsychology data was collected within 1 year of the MRI data. In addition, to reduce complexity of the statistical models the variables that coded for position along the anterior-posterior and inferior-superior directions in 20 positions were recoded to 10 positions each.

Statistical analyses were performed on these datasets. First, datasets were checked for outliers. Specifically, distances >1.5 times the interquartile range were removed by y and z position (1.8%). In addition, any MR session that had a Euler number (Rosen et al., 2018) smaller than -380 was also removed from the dataset (2.6%). The final dataset contained 616,460 datapoints and was analyzed using a mixed effect regression approach. The basic model we used was

$$\text{Distance} \sim \text{Sex} + \text{Euler} + \text{eTIV} + \text{Patient Status} + \text{Yposition} * \text{Zposition} * \text{Age} + \text{rand(MR session)}. \tag{1}$$

as well as an image of the Desikan-Killany atlas (aparc + aseg) in the native T1w space. All subsequent analyses were performed in native T1w space.

### Global distance analyses
Global distances were computed between points on the outer edges of the brain. Specifically, we placed 400 points on the outer cortical surface on the axial (horizontal) plane such that they were equally spaced along 20 locations in the inferior-superior (z) and 20 locations in the anterior-posterior (y) direction. The distance was then computed as the straight line between a point on the outer cortical ribbon

In this model we intended to test the (null) hypothesis that the effect of age on the distance between points in the brain was the same for all brain locations. At the same time, we controlled for the sex of the participant (male, female), the Euler number, the estimated total intracranial volume, and the patient status (CDR = 0, control; CDR > 0, patient). To account for the likelihood that these effects may vary between different MR sessions we included a random intercept for the MR session. All numerical variables were scaled such that they had zero mean.

In another model we examined the effect of the patient status while also controlling for age:

$$\text{distance} \sim \text{Sex} + \text{Euler} + \text{eTIV} + \text{Age} + \text{Yposition} * \text{Zposition} * \text{Patient Status} + \text{rand(MR session)}. \tag{2}$$

in the left hemisphere and a point on the outer cortical ribbon in the right hemisphere for each y,z pair (see Table 4 for pseudo code, note the medial wall was excluded). Conceptually, this is akin to slicing a sphere into 20 concentric circles and then computing the distance between points on the extreme left and right edges of the circle at 20 equally spaced positions in an anterior to posterior direction (see Fig. 1A for a visualization of this process).

These distances were computed using the T1w images that were aligned to the ACPC but were in native T1w space. We used the cortical ribbon file that was produced by Freesurfer v6.0 to compute the distances. We made sure the distance between points was computed when these points were in opposite hemispheres by consulting the label values present in the ribbon file. This resulted in a dataset that contained 400 distances between points for each MR session. Each distance was coded by its specific position in the brain in the y and z direction.

This dataset was then further augmented with demographic data (age, sex) as well as with other imaging, behavioural and

Finally, we examined the impact of three cognitive variables (episodic memory, working memory, and executive function) on the pattern of compression and expansion. Scores on these tasks were extracted from the OASIS3 datafiles and were collected using the Neuropsychological Battery Uniform Data Set (UDS)[24,55,56]. These neuropsychological tests were collected by the Alzheimer's Disease Resource Center (ADRC) and are used to assess clinical populations potentially experiencing cognitive impairment.

Episodic memory was assessed with the Logical Memory Test. Logical Memory consists of reading a short story to the participant and asking them to repeat the story back using as close to the same words as possible. The scores range from 0 (no recall) to 25 (complete recall). This is asked once right after telling the story (immediate recall) and another time about 30 min after the initial administration (delayed recall). This test allows us to examine episodic memory by examining the features of the story that the person is able to recall. We used the delayed recall score as our episodic memory measure for analyses.

Executive function was assessed with the Trail Making Test (TMT). In the TMT Part-A, the participant is shown numbers in circles spread throughout a piece of paper. The participant is asked to draw a line from one sequential circle to the next (1–2–3–4–5…, etc). In the TMT Part-B, the participant is shown various numbers and letters in circles spread throughout a sheet of paper, and asked to draw a line from one circle to the next, alternating in order between the numbers and letters (A-1-B-2-C-3… etc.). The participant is asked to stop if they make a mistake and begin from the circle before the mistake was made while the timer continues to run. The score for each part is determined by how long (in seconds) it takes the participant to complete connecting all of the circles. We created a composite z-score combining

### Table 4 | Pseudo-code for finding global distances

```
FOR each z in num_slices:
    CALCULATE y_values EQUALLY SPACED between min and max y-values in z-slice
    FOR each y in y_values:
        FIND min_x and max_x corresponding to y and z
        IF VALID POINTS found:
            distances[y, z] <- max_x - min_x
```

performance combining Part-A and Part-B performance. Because higher scores represent worse performance, unlike the other neuropsychological measures included in this study, effect directions were flipped for consistent interpretation of higher scores representing better cognition.

Finally, working memory was assessed with the Digit Span (DS) test. DS-Forward consists of reading a list of numbers to the participant then asking the participant to repeat the list of numbers back immediately after the administrator is finished. For DS-Backward, participants must repeat the numbers in reverse order. Scores for each are determined by how many trials they are able to repeat back correctly. We created a composite z-score combining performance on the DS-Forward and DS-Backward scores for analyses of working memory.

The model we used to examine the impact of these variables was:

effect modelling. We converted the output of these rather extensive regression tables into more comprehensive ANOVA tables with $p$-values using the Saitherwait approximation using the lmerTest package (v3.1-3). Whenever relevant interactions were significant ($p < 0.05$) post-hoc tests with bonferroni corrections for multiple comparisons were performed using the emmeans package (v1.8.4-1).

All effect sizes are reported as "z ratios", which is a statistic representing the effect size and is standard output from the "emmeans" package in R (beta value divided by standard error). The z-ratio can be interpreted as you would a $t$-statistic, however it contrasts the observed statistic against a normal distribution rather than against a $t$-distribution, which has a slightly different shape from a normal distribution.

$$\text{Distance} \sim \text{Sex} + \text{Euler} + \text{eTIV} + \text{Age} + \text{Patient Status} + \text{Yposition} * \text{Zposition} * \text{Cognitive Test} + \text{rand(MR session)}, \qquad (3)$$

where Cognitive Test indicated the scores for the Logical Memory, Executive Function or Working Memory test.

In all of these models our main effect of interest was in the triple interaction between Yposition, Zposition and the relevant variable (age, patient status, cognitive test). When this interaction was significant ($p < 0.05$), we performed post-hoc tests using the emmeans package[57]. Specifically, we tested the effect of the relevant variable for each combination of the levels of the Yposition and Zposition variables.

### Regional homologue analyses

The homologue distance was defined as the Euclidean distance between the centroid of the same region in the left and right hemisphere. Centroids were computed for all the cortical and subcortical regions in the left and right hemisphere of the Desikan-Killany atlas[58]. This resulted in a set of 39 distances between 6 subcortical and 33 cortical regions in either hemisphere for each MR session (see Supplementary Fig. 1 for list of regions). As before, these data were combined with demographic, imaging, and neuropsychology data collected at other occasions. Data were analyzed using comparable cleaning methods and statistical modelling techniques as before. Specifically, the basic regression model took the form:

### Sensitivity analyses

Increases and decreases in the number of points were performed by changing the number of points in the algorithm in the y and z direction outlined above. All other aspects remained constant. To perform the statistical analyses without reducing the number of factor levels we ran the exact same statistical model as outlined above. The bootstrap analyses were performed using the boot package (v1.3) in R with default options. Longitudinal analysis was performed by subsetting the dataset to those participants that were scanned more than once and selecting for each participant the initial and subsequent imaging session that maximized their distance in time. The resulting dataset was analyzed with the following model:

$$\text{Distance} \sim \text{Euler} + \text{eTIV} + \text{Yposition} * \text{Zposition} * \text{Time} + \text{rand(participant)}, \qquad (5)$$

where variables Euler, eTIV, Yposition and Zposition were defined as before. The variable Time had two levels (baseline and subsequent scanning session), and the random slope per participant captured potential variability of these effects within a participant.

For the analyses that relied on the classification of global distance measures as gyrus or sulcus, we consulted the Destrieux atlas for

$$\text{Distance} \sim \text{Sex} + \text{Euler} + \text{eTIV} + \text{volumeR1} + \text{volumeR2} + \text{Patient Status} + \text{Region} * \text{Age} + \text{rand(MR session)}, \qquad (4)$$

where variables sex, Euler, eTIV, patient status and age were defined as before. We were again mainly interested in the interaction between age and region that would test whether age affected the distance between regions. In addition to the models tested above, we also controlled for the volume of the left (volumeR1) and right (volumeR2) regions. When the region by age interaction was significant, we performed post-hoc tests using the emmeans package[57]. Specifically we examined the effect of age for each region. As before, we also examined how patient status and cognitive test interacted with the region variable.

In all these models our main interest was in the interaction term that tested whether a particular variable of interest (age,

obtaining the appropriate label (Freesurfer file aparc.a2009s + aseg). Analyses performed on the Cam-CAN dataset were restricted to gyrus-to-gyrus distances and relied on the same model formula as for the unrestricted data. To assess deviation from a straight line, we quantified how much homologue pairs diverged while holding the X coordinate fixed. Specifically, we computed the deviation in the Y and Z coordinates as the absolute difference between the respective coordinates of each homologue pair. The total deviation was then obtained by summing the deviations in Y and Z. This measure provided an index of the extent to which each homologue pair deviated from a straight line within each MR session. We then tested whether total deviation impacted the critical Region by Age interaction in the following model:

$$\text{Distance} \sim \text{Sex} + \text{Euler} + \text{eTIV} + \text{volumeR1} + \text{volumeR2} + \text{Patient Status} + \text{Total Deviation} * \text{Region} * \text{Age} + \text{rand(MR session)}, \qquad (6)$$

patient status, cognitive test) affected the distance between points at different locations in the brain. All statistical modelling was performed in R (v4.1.2) using the package lme4 (v1.1-31) for mixed

where variables were defined as before. Finally, analyses of data whose total deviation <10 used the same model formula as for the unrestricted data.

## Cam-CAN replication

We performed a subset of the analyses reported above on a dataset that was collected independently from the first dataset. Specifically, we preprocessed the Cam-CAN dataset with the same preprocessing pipeline as the OASIS3 test dataset. We then perform the exact same analyses on this validation dataset as on the test dataset with the exception of the effect of Patient Status (all participants were healthy elderly participants). We examined the impact of fluid intelligence on the patterns of compression and expansion. Scores were extracted from the Cam-CAN data files and collected using the Cattell Culture Fair test[25]. In this task, participants are shown four patterns at a time and are instructed to select the "odd one out." The difficulty between these puzzles vary from being easy to difficult. This captures fluid intelligence by reflecting the mental control used by complex activities.

## Reporting summary

Further information on research design is available in the Nature Portfolio Reporting Summary linked to this article.

## Data availability

The raw OASIS3 and Cam-CAN data are protected and are not available due to data privacy laws. The processed OASIS3 and Cam-CAN data used in this study are available in the CodeOcean database [https://doi.org/10.24433/CO.3593979.v1].

## Code availability

The code generated during the current study is available on Code Ocean [https://doi.org/10.24433/CO.3593979.v1].

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

## Acknowledgements

This work was supported by Grant PID2021-127611NB-I00 (to N.J.) funded by MICIU/AEI/10.13039/501100011033 and by the European Union, and supported by the National Institute of Health grants R01AG053555 (to M.A.Y.) and F32AG074621 (to J.N.A). Data were provided by OASIS-3: Longitudinal Multimodal Neuroimaging: Principal Investigators: T. Benzinger, D. Marcus, J. Morris; NIH P30 AG066444, P50 AG00561, P30 NS09857781, P01 AG026276, P01 AG003991, R01 AG043434, UL1 TR000448, R01 EB009352. Data collection and sharing for this project was also provided by the Cambridge Centre for Ageing and Neuroscience (CamCAN). CamCAN funding was provided by the UK Biotechnology and Biological Sciences Research Council (grant number BB/H008217/1), together with support from the UK Medical Research Council and University of Cambridge, UK.

## Author contributions

Conceptualization, all authors; Methodology and Software, N.J.; Formal Analysis, N.J., J.N.A., Y.Y.E.; Writing—Original Draft, Y.Y.E., J.N.A., M.Y., N.J.; Writing—Reviewing and Editing, all authors.

## Competing interests

M.A.Y. is Co-founder and Chief Scientific Officer of Augnition Labs, L.L.C. All other authors declare no competing interests.
