## [Transparent Peer Review file · Nature Communications]

Age-related constraints on the spatial geometry of the brain

Corresponding Author: Dr Niels Janssen

Version 0:

Reviewer comments:

Reviewer #1

(Remarks to the Author)

This paper investigated how the spatial anatomy of the brain changes across age. The authors found that age is associated with brain shape expansion in inferior-anterior regions and compression in superior-posterior regions. They also found that these shape changes are associated with clinical and cognitive-related functioning. In most respects, I found the paper to be interesting. However, I do have concerns/issues, which are itemized below.

1. My most major concern is on the measurement employed. Equidistant points were placed in the left and right hemisphere, which were then connected by a straight line to calculate the Euclidean distance. This is problematic with respect to the following issues, especially because the authors are performing their analyses in native space:

a.) The study is cross-sectional such that Euclidean distances have person-specific reference values that are not generalizable across the cohort. Hence, it is difficult to ascertain whether shape difference, e.g., between age 30 and 40, is due to real expansion/compression or that the subject at age 40 just naturally has a larger/smaller brain. The authors should perform a supplementary analyses using longitudinal data, such as from the same OASIS dataset, to ensure that this issue does not affect their conclusions.

b.) Because brain sizes across individuals are different (some people may have elongated brain along the anterior-posterior axis), how can the process of placing equidistant points be consistent across individuals? Elongated brains will have larger spacing between points compared to shorter brains.

c.) Placing of points do not respect the intricate gyrfication of the brain, such that left/right point could be in a sulcus and the bilateral point could either be in a sulcus or gyrus. This effect biases the distance measure.

2. What was the rationale behind using 20 locations (400 points)? Are results robust against this number?

3. Following Comment 2, it is unclear why there is a need to reduce the number of locations from 20 to 10 in the analyses. The argument was to reduce the complexity, but I don't see why that's a problem in their statistical models. It is also unclear how the 10 positions were chosen and whether changing the number of positions will affect the results.

4. In some instances, the authors related their findings to regional gray matter changes. This is not accurate as the shape changes measured are not directly related to atrophy. Different mechanisms could be in play such as unihemispheric physical changes, changes in gray matter, changes in white matter, changes in ventricles, and allocation of points. The authors need to revisit their discussion to avoid making these inaccurate links to current literature.

5. Finally, given the issues raised in Comments 1 and 4, I'm struggling to identify the benefit of using the measure proposed in this study. More importantly, the study finds similar patterns found in the literature, but fails to provide a mechanistic understanding of why the expansions/compressions happen in the first place. This is crucial to distinguish whether the novelty of the study is only the measure proposed or also the resulting findings.

6. Computer codes need to be provided for replication.

(Remarks on code availability)

The code provided only has one script to perform an example analysis given a nifti input. However, no codes were provided to reproduce the results of the paper. There are also no README files to explain the input needed for the code provided.

Reviewer #2

(Remarks to the Author)

The authors present a study on measures of cortical expansion and compression in the aging brain. Using distance measures, they were able to show relevant gradients across anterior-posterior and superior-inferior axes mainly related to compression associated with age, but also with clinical status and cognitive performance.

The study is based on a very large dataset as test and an independent, also reasonably large dataset as replication dataset. The overall approach is interesting and might provide new markers for measuring global, but also regional changes of the brain.

I nevertheless have some comments to be considered further:

- The results of the global and regional analyses seem to differ in that for the global analysis, age, clinical status and cognition seem to exhibit similar effects, with compression being a dominant feature, associated with partial expansions (Figs. 2/3; while for the regional analysis, the dominant effect of age seems to be expansion (Fig. 4), but for clinical status and cognition, it seems to be compression (Fig 5). How could that be interpreted?
- Related to that and to the interpretation in the Discussion the authors provided on a potential 'slouching' phenomenon which might explain the observed expansion in particularly inferior/anterior parts of the brain – would it be possible that due to the close proximity to the skull base particularly in these regions some methodical difficulties, i.e. susceptibility effects, known for T1-weighted imaging could have caused such effects?
- The results of the global and regional analysis are not easy to compare, given the different depictions in the figures (Figs 2/3 with slice stacks vs. Figs. 4/5 with schematic surface views). I would recommend to change the depictions in Figs. 2/3 to the versions used in Figs. 4/5 as this is way more intuitive.
- The two datasets could not be used for a replication of all the investigated effects, due to differences in sample characteristics and neuropsychological tests applied. While I agree that a replication in a sample with somewhat different settings can speak to the robustness of the method and results, a more direct replication particularly of the cognitive effects, but also the clinical status effects would be valuable. This could be achieved by either including yet another dataset which has the relevant data or via an internal validation of the results, i.e. either via a sample split of OASIS in discovery and test or via a, e.g., bootstrap approach testing for stability of effects with random left-out parts of the sample. I think this would further strengthen the conclusions of the manuscript.
- In the extended data figures 1 & 2, effect sizes are reported, but in figure 1 with the label 'z ratio' on the x-axis, in figure 2 with the label 'effect size'. This should be made consistent and explicit – what is meant by 'effect size' in figure 2 exactly, how was it measured? And how is a z ratio reflecting an effect size (as in figure 1)?

(Remarks on code availability)

Reviewer #3

(Remarks to the Author)

Thank you for the opportunity to review the manuscript of Escalante and colleagues.

The study examines age-related neuroanatomical changes as linked geometric patterns. Accordingly, the study quantified age-related neurostructural changes by measuring global expansion and compression, as well as distances between homologous regions in structural MRIs from 1059 individuals from the OASIS study (discovery sample) and 642 individuals from a replication sample (CamCAN). The study found that aging was associated with global shape expansion in inferior-anterior gradients, compression in superior-posterior gradients, and regional expansion between frontotemporal homologues. These patterns were linked to clinical impairment and cognitive deficits

The study has the following strengths

1. The issue of how best to capture age-related brain changes is topical and currently popular
2. Substantial sample size of the discovery (OASIS) and independent validation datasets (CamCAN)
3. Open science framework – i.e., use of public dataset – code availability for review
4. Age range of the sample covers most of middle and late adulthood
5. Use of established pre-processing pipelines as per the human connectome project
6. Innovative approach to the study of the effect of age on the spatial patterns of neuroanatomy.
7. Examination of the functional significance of the neuroimaging metrics

Areas for revision/clarification

1. The OASIS sample includes longitudinal data – could the authors consider a pilot analysis to test whether the population level age-related changes also reflect intra-individual trajectories?
2. Global distances: please provide some justification for the number of points used – was it random? How does this number affect results
3. Global distances: The way this is explained in the method is that the axis considered for expansion-compression was along the sagittal plane (i.e., distances were calculated between a point on the cortical ribbon of the left hemisphere and the homologous point on the right hemisphere). Please justify and if this comment is incorrect please expand and clarify.
4. The choices for the basic model require further clarification – why no sex-specific analysis instead of modeling sex? Or perhaps both, the clarify the role of sex as it is a major moderator of brain aging
5. On the same topic, why the inclusion of the total intracranial volume – could the authors produce any evidence that this is correlated with distances between points? This is intriguing as the association between volume and geometry at least in regional brain anatomy is not widely examined or known
6. Could the authors comment how their data compare with other global measures of aging such as brain-age-gap-estimation? What aspect of the expansion/compression is associate with other biological meaningful measures of aging?

(Remarks on code availability)

Looking at the script, I would have liked to see more detailed instructions

Version 1:

Reviewer comments:

Reviewer #1

(Remarks to the Author)

The authors have done a great job in addressing my (Reviewer 1) and other Reviewers' previous comments/suggestions. The paper is now much more improved; however, I have residual comments, which are itemized below.

1. In relation to my Comment 1c, I meant that a point on the left hemisphere could be on a gyrus whilst the corresponding point on the right hemisphere could be on a sulcus or vice versa. Distances could significantly differ when measured from gyrus-to-gyrus, gyrus-to-sulcus, or sulcus-to-sulcus. Hence, on cross-sectional data like the one used in the study, for the same point of interest (e.g., point at the far end of anterior-inferior axis), a hypothetical scenario of one subject having a gyrus-to-gyrus distance vs another with a sulcus-to-sulcus distance will lead to a biased conclusion of compression. Reducing the number of points from 20 to 10 does not resolve this. Hence, either show that this effect is relatively minor and doesn't change the conclusions of the study or highlight potential specific effects of between-subject folding variations in the Discussion.

2. In relation to my Comments 2 and 3, I disagree with the authors' statement in the revised text that the spatial patterns are generally the same, which is not the case especially towards the superior direction when the number of locations was reduced from 20 to 15. The fact that the differences are clearly visible brings back my concern about the robustness of the results. The authors should also ensure that robustness is not only tested by visual inspection of spatial patterns but also on all statistical analyses of associations with clinical and cognitive variables.

3. There is also another issue with the calculation of Euclidean distance between regional homologues. The distance was measured between the center of masses, but homologous regions are not perfectly symmetric and could have differences in their y- or z-axis locations; hence, connecting a straight line will not be perfectly horizontal. This can also bias the expansion/compression conclusions because the hypotenuse of a right triangle is longer than the base. This bias could be the reason why results in Figures 2 and 4 are largely inconsistent.

(Remarks on code availability)

There is no README file explaining what the folders and scripts represent.

Reviewer #2

(Remarks to the Author)

The authors adequately addressed all points previously raised. I particularly value the efforts the authors put in creating such comprehensive videos which facilitates interpretation of the results pretty well. I do not have any further comments or concerns.

(Remarks on code availability)

Reviewer #3

(Remarks to the Author)

The authors have been very responsive to the reviewers' comments

(Remarks on code availability)

Version 2:

Reviewer comments:

Reviewer #1

(Remarks to the Author)

The authors have done a great job in sufficiently addressing my previous residual comments. I appreciate the authors for performing several additional analyses to ensure robustness, which are important because the method they're proposing is something new to the field. I commend them for the scientific rigour they presented. I am happy to recommend the publication of the work.

(Remarks on code availability)

We thank the Reviewers for their insightful comments and for the opportunity to revise our manuscript. We believe the additional analyses performed and changes to the manuscript from the Reviewer's suggestions resulted in a greatly strengthened manuscript that is of quality for publication in *Nature Communications*. We particularly addressed concerns of measurement parameters, internal validity, and the cross-sectional design with a number of new analyses, presented in detail below, which demonstrate our results are robust to methodological design. Additionally, we paid careful attention to the concern raised by Reviewer 1 and have now clarified the novelty and value of the study to understanding how aging affects the spatial geometry of the brain, which has a number of critical clinical and basic neuroscience implications (see full response to Reviewer 1, comment 5).

We have provided point-by-point responses to each comment raised by the Reviewers. Changes to the text and additional analyses/figures are replicated below each comment for ease of review. Within the revised manuscript and this response document, changes are indicated by **highlighted text**. The revised manuscript has been formatted in accordance with the guidelines of *Nature Communications*.

REVIEWER COMMENTS

Reviewer #1 (Remarks to the Author):

This paper investigated how the spatial anatomy of the brain changes across age. The authors found that age is associated with brain shape expansion in inferior-anterior regions and compression in superior-posterior regions. They also found that these shape changes are associated with clinical and cognitive-related functioning. In most respects, I found the paper to be interesting. However, I do have concerns/issues, which are itemized below.

1. My most major concern is on the measurement employed. Equidistant points were placed in the left and right hemisphere, which were then connected by a straight line to calculate the Euclidean distance. This is problematic with respect to the following issues, especially because the authors are performing their analyses in native space:

a.) The study is cross-sectional such that Euclidean distances have person-specific reference values that are not generalizable across the cohort. Hence, it is difficult to ascertain whether shape difference, e.g., between age 30 and 40, is due to real expansion/compression or that the subject at age 40 just naturally has a larger/smaller brain. The authors should perform a supplementary analyses using longitudinal data, such as from the same OASIS dataset, to ensure that this issue does not affect their conclusions.

We agree with the reviewer that within-subject, longitudinal analyses would be the most robust to potential sources of variation across participants. Because our study was interested in long-term brain structural change (over decades), we initially performed this with cross-sectional data. However, we have now additionally performed longitudinal analyses within the OASIS sample at the suggestion of the reviewer to verify the robustness of our cross-sectional effects. While the follow-up time between scans in OASIS was much shorter in duration (at baseline mean age in yrs = 68.5; SD = 9.6; at follow-up mean 73.5; SD = 8.5) compared to the broad age range we examined cross-sectionally (30-90+ years), and reduced in statistical power due to the smaller number of longitudinal time points available (N = 499 participants with longitudinal data), we were still able to largely replicate our results (see **Supplementary Figure 6**, shown below).

For this exploratory analysis, we analyzed all participants with at least 2 timepoints of structural scans and maximized the time between their baseline and follow-up scan, resulting in 499 participants included in the

analysis. We observed a significant effect of time on gradients of expansion and compression across both global distances ($F(81) = 2.68, p < 0.0001$) and regional homologues ($F(38) = 4.03, p < 0.0001$). The patterns of expansion and compression for both analyses are generally consistent with our cross-sectional results, although a bit weaker due to a reduction in statistical power and dramatically shorter follow-up time. For the global distance results (panel **A** below), we find a similar inferior-anterior pattern of expansion, and posterior pattern of compression. For the regional homologue results (panel **B** below), we primarily observe expansion between subcortical regional pairs, such as caudate, thalamus, and amygdala, which were also the regions that demonstrated the strongest expansion pattern in our cross-sectional results. We present the results both corrected and uncorrected for multiple comparisons to better represent the overall pattern of expansion and compression to compare with the cross-sectional results. We believe that even with concerns about statistical power and the much shorter follow-up time (years rather than decades), this analysis strengthens the conclusions in our cross-sectional analyses.

We have added these new results as a sensitivity analysis (see the new “*Sensitivity Analyses*” results section) and include these results as a new figure (**Supplementary Figure 6**).

*Finally, we attempted to further validate our cross-sectional results by exploiting the longitudinal component of the OASIS3 dataset. Post-hoc construction of a longitudinal dataset by maximizing the number of years between baseline and follow-up scans yielded 499 individuals whose mean age was 68.5 years old (SD = 9.6 years) at baseline and 73.5 years old (SD = 8.5 years) at follow-up. We performed an analysis modeling longitudinal change in global distance and regional homologue expansion and compression to further validate our cross-sectional results. The results were in line with our earlier observations for both global distance ($F(81,309996) = 2.68, p < 0.0001$) and homologues ($F(38,37260) = 4.03, p < 0.0001$; see **Supplementary Figure 6**). Future longitudinal work with increased time between timepoints should further examine this issue.*

Supplementary Figure 6. Sensitivity analyses demonstrating exploratory longitudinal within-subject change in brain geometry. Within OASIS3, we performed post-hoc construction of a longitudinal dataset by maximizing the number of years between baseline and follow-up scans yielded 499 individuals whose mean age was 68.5 years old (SD = 9.6 years) at baseline and 73.5 years old (SD = 8.5 years) at follow-up. Models of longitudinal change in **(A)** global distance and **(B)** regional homologue expansion and compression were in line with cross-sectional results. Results are presented both corrected (pFDR<0.05) and uncorrected (punc<0.05) for multiple comparisons to better represent the overall pattern of expansion and compression to compare with the cross-sectional results.

Future studies applied to datasets with greater within-subject follow-up time will be necessary to definitely show the cross-sectional pattern of results and relate these to concurrent cognitive changes. We discuss this further in the Limitations section of the Discussion, shown below:

Finally, we were not powered to conduct a robust longitudinal assessment of within-subject change in expansion and compression patterns alongside concurrent change in clinical status and/or cognition. Future studies should determine the temporal trajectory of structural versus cognitive changes to better assess the predictive value of imbalances between expansion and compression.

b.) Because brain sizes across individuals are different (some people may have elongated brain along the anterior-posterior axis), how can the process of placing equidistant points be consistent across individuals? Elongated brains will have larger spacing between points compared to shorter brains.

In our analyses, the placing of points takes place in native space. Given that head shape generally varies between participants the reviewer is correct in assuming that the spacing between these points is not exactly the same across individuals. However, two important aspects should be mentioned. First, equidistant spacing between points across individuals can only be achieved when brains are normalized. However, normalization of brains is not desirable for our current study since it would remove any variability in head shape and therefore prevent examination of how brain shape changes with age. Second, given that the locations are always spaced

evenly throughout the brain within a given individual, the *relative location* of a given set of points is generally consistent across individuals. For example, consider two individuals, one with a very elongated shape in the Z (inferior-superior) direction and one with a less elongated shape in this direction. When we place 20 Z locations in both individuals, even though the spacing between successive locations differs between these individuals, the locations generally capture the same location in the brain. In other words, consistent spacing can only be achieved with normalization, which is not desirable, and because locations are always evenly spaced throughout the brain, spacing is relatively consistent across individuals. We thank the reviewer for highlighting this important aspect of our analyses.

c.) Placing of points do not respect the intricate gyrification of the brain, such that left/right point could be in a sulcus and the bilateral point could either be in a sulcus or gyrus. This effect biases the distance measure.

We thank the reviewer for raising this interesting point. If we understand correctly, the reviewer is arguing that our distance measure may be noisy because at one location the two lateral points may be in a sulcus, while at the next location down the two lateral points may be on a gyrus. We think that at heart this is a question about how to measure the distance (or width) of an object that has an irregular (folded) shape. While we think that this is an issue that is certainly not resolved in a straightforward manner, we also think that our current solution is the simplest one. Moreover, our statistical analyses mitigate the impact of such noise because while we collect 20 x 20 points on the brain's surface, we group these points into 10 x 10 factor levels. In this way we are effectively averaging out such noise as described by the reviewer. In addition, note that as we report in our supplementary sensitivity analyses, that with increasing points, detection of the critical interaction was more reliable.

We have now mentioned that variability in gyrification may be a limitation of our results in the Discussion, shown below:

Factors such as the individual variability in head size and shape or gyrification patterns may have influenced our results, since analyses were performed in native space as necessary with this method.

2. What was the rationale behind using 20 locations (400 points)? Are results robust against this number?

We initially chose 20 locations (400 points) to sufficiently cover the anterior to posterior and inferior to superior gradients without too sparse of sampling (less points) or to become too computationally expensive to process (more points). To demonstrate robustness against this methodological decision, we have reperformed the analysis using 15 locations (225 points) and 30 locations (900 points). Regardless of increasing or decreasing the number of locations, we are able to replicate our results, finding patterns of significant inferior-anterior expansion and superior-posterior compression (15 locations: $F(81) = 10.03$, $p < 0.0001$; 30 locations: $F(81) = 37.72$, $p < 0.0001$). This has now been added as a sensitivity analyses (see the new "Sensitivity Analyses" Results section) and provided as **Supplementary Figure 4A**, replicated below:

First, to determine if the number of points (20 x 20 locations, 400 points) placed for global distance analysis influenced observed results, we repeated analyses with a smaller (15 x 15 locations, 225 points) and greater (30 x 30 locations, 900 points) number of points along the outer edge of the brain. The association between global shape change and age remained significant regardless of the

number of points (225 points: $F(81,324968) = 10.03$, $p < 0.0001$; 900 points: $F(81,1455126) = 37.72$, $p < 0.0001$), and spatial gradients of expansion and compression were generally consistent with the original results (**Supplementary Figure 4A**).

Supplementary Figure 4. Sensitivity analyses varying methodological parameters for the global distance analyses. (A) Patterns of global expansion and compression were similar to initial results (20 x 20 locations, 400 points) when varying the number of points placed along the outer edge of the brain to a smaller (15 x 15 locations, 225 points) or greater (30 x 30 locations, 900 points) number.

3. Following Comment 2, it is unclear why there is a need to reduce the number of locations from 20 to 10 in the analyses. The argument was to reduce the complexity, but I don't see why that's a problem in their statistical models. It is also unclear how the 10 positions were chosen and whether changing the number of positions will affect the results.

In the statistical analysis, the number of points was reduced to 10 to reduce computational RAM requirements of the model. We have now repeated analyses with 20 points, which was statistically significant ($F(361) = 18.51$, $p < 0.0001$) and replicated the original pattern of inferior-anterior expansion and superior-posterior compression based upon 10 points (see **Supplementary Figure 4B** below). While the 20 point statistical analyses was perhaps even more robust statistically, we have kept the reduction to 10 points in the main manuscript and in our open-access code to allow the code to be run on a greater variety of computers with varying RAM loads (the 20 point model required 80GB of RAM).

The replication with 20 points is now added as a new analysis (see the new “Sensitivity Analyses” Results section, replicated below) and provided in **Supplementary Figure 4B**, shown below:

Next, we determined if downsampling the number of points during statistical analyses (20 points to 10 points to facilitate analytic computations) influenced our results. With no downsampling in the statistical models, we observed convergent results with our original analyses ($F(361,1454519) = 18.52$, $p < 0.0001$; **Supplementary Figure 4B).**

Supplementary Figure 4. Sensitivity analyses varying methodological parameters for the global distance analyses. (B) With no downsampling (reducing 20 points to 10 points to facilitate analytic computations) applied in the statistical models, we observed convergent results with our original analyses.

4. In some instances, the authors related their findings to regional gray matter changes. This is not accurate as the shape changes measured are not directly related to atrophy. Different mechanisms could be in play such as unihemispheric physical changes, changes in gray matter, changes in white matter, changes in ventricles, and allocation of points. The authors need to revisit their discussion to avoid making these inaccurate links to current literature.

We agree that comparing our results to regional gray matter changes is not a one-to-one comparison, as regional gray matter changes reflect isolated atrophy, as our compression and expansion results reflect changes to the spatial relationships between regions or overall brain geometry. We have revised our comparison to previous regional results, reproduced below:

Our findings provide novel insights that build upon existing literature examining structural changes in the brain during the second half of aging by applying a unique approach to quantify gradients of change of whole brain shape, representing the inferior-superior and anterior-posterior axis of the brain. Previous work characterizing gross anatomical changes of the aging brain have largely focused on neurodegenerative or volumetric changes to regions or lobes in isolation. Although these studies highlight reduced volume in age-vulnerable regions such as the hippocampus, subcortical structures, and prefrontal cortex, this work does not characterize the relative change in position between these structures. In contrast, we demonstrate that these same structures are prone to relative spatial changes, with expansion in inferior-anterior and compression in superior-posterior aspects of the brain. Expansion predominantly targeted inferior-anterior regions such as the medial temporal lobe and subcortical regions. In contrast, compression was observed in superior-posterior aspects of the brain.

We also agree with the reviewer that a variety of mechanisms may contribute to our results. We have updated the Discussion as suggested to include these alternatives. Updated text is shown below:

One limitation of interpretation of results within this study is that a number of mechanistic contributors could have led to the observed patterns of whole brain and regional expansion and compression. Although our analyses ruled out that ventricular expansion, total intracranial volume, or regional atrophy accounted for the observed expansion effects, other biological changes may impact geometric brain shape changes, such as unihemispheric brain changes or specific changes to white or gray matter.

5. Finally, given the issues raised in Comments 1 and 4, I'm struggling to identify the benefit of using the measure proposed in this study. More importantly, the study finds similar patterns found in the

literature, but fails to provide a mechanistic understanding of why the expansions/compressions happen in the first place. This is crucial to distinguish whether the novelty of the study is only the measure proposed or also the resulting findings.

We believe our results represent not only a methodological contribution, but a new perspective of how to study age-related structural brain changes by considering spatial geometry. The spatial geometry of the brain has recently emerged as a key component that influences how the brain processes information. In this light, considering overall geometric changes to the shape of the brain as a function of aging could provide new insights into brain aging and age-related cognitive changes, though this has never been considered in past research. While our results implicate many of the same brain regions (e.g. frontotemporal cortex) shown to undergo isolated regional brain atrophy, the current study greatly expands knowledge by showing that overall spatial relationships, even when controlling for regional atrophy, change with age and are associated with clinical impairment and cognitive performance in older adults. This suggests that factors that have been previously unexplored, perhaps mechanical forces or a breakdown in structural scaffolding (elaborated on in the Discussion) may contribute to cognitive deficits in aging over and above isolated volumetric changes, providing an exciting new direction for the field to further explore the mechanistic basis behind.

We now further emphasize the importance of considering spatial relationships between regions rather than isolated volumetric changes in the Introduction:

While these previous methods account for how the volume of each region or deformation of each voxel changes in *isolation* in respect to aging, these standard approaches are not designed to characterize complex, simultaneous changes between the spatial relationships across regions. However, understanding how aging affects geometric features such as the global shape of the brain or spatial distances between regions would provide critical insight into the overall anatomical reorganization of the brain with age. This is particularly important to characterize because changes to the complex spatial anatomy and geometry of the brain may impact the ability of the brain to properly function and process information^{19,20}. Recent work has shown that the shape of the cortex confers geometric constraints to the functional dynamics of the brain, with a greater impact on activity than that of structural connectomes representing white matter²¹. To our knowledge, this framework of examining changes in the global geometry of the brain has not been applied to the study of age-related changes in brain structure. Critically, if the spatial geometry of the brain in fact impacts functional dynamics²¹, these age-related changes to the spatial geometry of the brain may be one underlying mechanism associated with clinical impairment and age-related cognitive changes.

We now also further emphasize the implications of these results in the Discussion:

These findings have a number of implications for the study of age-related structural changes. First, they reinforce that considering the global shape and spatial geometry of the brain provides complementary information to the study of regional atrophy. Our results also may provide a basis for previously observed age-related changes in compression versus expansion in skull shape³⁰, which are thought to result from mechanical forces of the brain³¹. Finally, we demonstrate these global patterns of expansion and compression have cognitive implications even when controlling for age and regional atrophy patterns, in which the spatial geometry of the brain may strongly constrain brain function.

We tested multiple potential mechanistic effects that may lead to this pattern of results, such as ventricular enlargement, which could only partially explain these effects. In addition, note that both white and gray matter volume are known to reduce with age and hence, one may expect compression of the brain in temporal regions. Given that expansion was observed in temporal regions, it must be the case that other factors lead to

the observed changes with age. However, a more precise mechanistic account of why these changes are occurring is beyond the scope of the current study. Studies showing regional atrophy changes similarly do not provide a molecular basis for volumetric changes, which may be due to a number of microstructural properties such as synaptic loss, demyelination, and other molecular contributors. However, we plan to follow this line of research in future datasets with more measures providing potential mechanistic insight. This limitation, and the future directions of investigation, have been updated in the Discussion as shown below:

One limitation of interpretation of results within this study is that a number of mechanistic contributors could have led to the observed patterns of whole brain and regional expansion and compression. Although our analyses ruled out that ventricular expansion, total intracranial volume, or regional atrophy accounted for the observed expansion effects, other biological changes may impact geometric brain shape changes, such as unihemispheric brain changes or specific changes to white or gray matter. Therefore, future directions include the examination of how features such as white matter structural connections between regions^{41,42}, regional microstructural properties such as demyelination⁴³, vascular integrity^{44,45}, synaptic loss^{46,47}, and other molecular contributors that affect structural scaffolding in the brain may contribute to our findings.

While we cannot provide an exact mechanistic explanation to the expansion or compression effects, we do show that changes to the spatial geometry of the brain is associated with clinical impairment and cognitive performance, providing a potential mechanism for cognitive decline. Recent findings have suggested that the overall spatial geometry of the brain plays a fundamental role in shaping the efficiency of neural processing. Due to these findings, it is suggested that the overall shape of the brain may strongly contribute to deficits in cognition. We demonstrate this effect - expansion/compression in brain shape was associated with clinical impairment, memory, executive function, and working memory in spatially specific patterns that map onto known regions involved in these processes. Importantly, these results controlled for regional volume and ventricle size as well, suggesting that this effect cannot be fully attributed to ventricular enlargement or isolated neurodegeneration. This is a very novel finding that provides a potential new mechanism of cognitive dysfunction in aging. It also prompts the creation of reference curves of expansion and compression in line with Brain Age Gap Estimation models, suggested by **Reviewer 3**, which has now been added to the Discussion as an important future extension of this work:

The imbalance between inferior-anterior expansion and parietal compression may be a novel phenotype of clinical impairment. For example, creation of reference curves of global compression and expansion patterns associated with “normal” aging, such as done with Brain Age Gap Estimation models^{38,40}, could be created from much larger datasets and used to identify individual participant’s patterns of compression or expansion that may be related to known neurodegenerative disorders or clinical impairment.

Together, our study provides novel insights on structural changes to the aging brain that do not simply replicate previous work, but expand it by considering the geometric structure of the brain and spatial relationships between regions, which are associated with older age, clinical impairment, and cognitive performance. This provides critical information for the study of aging as well as methodological development.

6. Computer codes need to be provided for replication.

All code needed to reproduce these results and figures are now included in the Code Ocean facility.

Reviewer #1 (Remarks on code availability):

The code provided only has one script to perform an example analysis given a nifti input. However, no codes were provided to reproduce the results of the paper. There are also no README files to explain the input needed for the code provided.

All code needed to reproduce these results and figures are now included in the Code Ocean facility, including README files.

Reviewer #2 (Remarks to the Author):

The authors present a study on measures of cortical expansion and compression in the aging brain. Using distance measures, they were able to show relevant gradients across anterior-posterior and superior-inferior axes mainly related to compression associated with age, but also with clinical status and cognitive performance.

The study is based on a very large dataset as test and an independent, also reasonably large dataset as replication dataset. The overall approach is interesting and might provide new markers for measuring global, but also regional changes of the brain.

I nevertheless have some comments to be considered further:

1. The results of the global and regional analyses seem to differ in that for the global analysis, age, clinical status and cognition seem to exhibit similar effects, with compression being a dominant feature, associated with partial expansions (Figs. 2/3; while for the regional analysis, the dominant effect of age seems to be expansion (Fig. 4), but for clinical status and cognition, it seems to be compression (Fig 5). How could that be interpreted?

We believe our new 3D video animations of the global distance results (see Comment #3) help show the similarity of results between global distances and regional homologues now that they are visualized in a similar manner. The overall results of the global distance and regional homologue results were fairly consistent and can be interpreted together to suggest that there is relative expansion in inferior-anterior regions, which includes the medial frontal and temporal lobes. The pattern of compression, as the reviewer notes, is found for aging when looking at global distances, but not regional homologues. However, we do identify regional homologue compression as a unique feature of clinical impairment and lower performance on various cognitive domains. This suggests that compression may be uniquely associated with more severe clinical presentations and serve as an intriguing biomarker. This point is elaborated further in the Discussion section.

2. Related to that and to the interpretation in the Discussion the authors provided on a potential ‘slouching’ phenomenon which might explain the observed expansion in particularly inferior/anterior parts of the brain – would it be possible that due to the close proximity to the skull base particularly in these regions some methodical difficulties, i.e. susceptibility effects, known for T1-weighted imaging could have caused such effects?

We thank the reviewer for highlighting this aspect of our study. The reviewer is correct in stating that susceptibility artifacts affect T1w images and that such artifacts typically occur in areas of air-tissue interfaces such as in the base of the skull in inferior temporal regions. At the outset we should note that we attempted to mitigate the impact of susceptibility artifacts by using field map corrections and excluding images with large negative Euler numbers (fieldmaps were only available for the OASIS3 dataset). In addition, we find that the data we observed are not compatible with an explanation in terms of susceptibility artifacts. First, note that in

our homologue data, the strongest expansion effects were observed for the caudate nucleus (even after controlling for ventricle volume). Susceptibility artifacts are typically found in inferior temporal regions at the base of the skull. It is not obvious how susceptibility artifacts could account for expansion effects of regions much more superior to this portion of the brain. Second, given that our OASIS3 data was fieldmap corrected and the CAMCAN dataset was not, if susceptibility artifacts were driving our effects, one might have expected differences between the OASIS3 and CamCan datasets. However, both datasets showed very similar results. Finally, more generally, while susceptibility artifacts are known to cause signal dropout and geometric distortions, it is not clear how they would systematically increase the distance between contralateral points in inferior temporal regions with age, clinical status and cognition. In other words, although we cannot rule out some contribution of susceptibility artifacts to our observed effects, we believe such an impact to be limited (perhaps to inferior temporal regions), and that the distance effects we observed are more likely to have a biological cause. We have added this potential limitation to the Discussion:

A second potential concern relates to susceptibility artifacts on T1-weighted MRI that may primarily affect inferior-anterior aspects of the brain due to the close proximity to the skull base. This may have influenced our results, although linking such artifacts to changes in age, clinical status and cognition is not obvious and would not predict the observed expansion effects in brain regions relatively far removed from the skull base (i.e., caudate, thalamus).

3. The results of the global and regional analysis are not easy to compare, given the different depictions in the figures (Figs 2/3 with slice stacks vs. Figs. 4/5 with schematic surface views). I would recommend to change the depictions in Figs. 2/3 to the versions used in Figs. 4/5 as this is way more intuitive.

We thank the reviewer for this suggestion. We have now worked hard to create an equivalent visualization to the “ggseg” produced figures in Figures 4/5 - we could not use the ggseg package here as these points are along the exterior of the brain, rather than volumetric ROIs.

We have created 2D visualizations of the points plotted on a brain, which we added to Figure 2 in addition to the original slice views. We also created a corresponding 3D video with the visualization rotating to fully capture the compression and expansion gradients. The updated version of Figure 2 (see **Fig 2C**) is reproduced below, and the 3D video file is now supplied as **Supplementary Video 1**. We have created a similar 3D video for the age group results (**Fig 2B**), now added as **Supplementary Video 2**. We believe these new visualizations provide a helpful representation of our global distance results.

Figure 2. Age-related changes to global distances between the outer edges of the brain. Effects of age were assessed continuously (**A**) and by comparing discrete groups of increasing age (Groups 2-8) to a young control group (Group 1) to detect cross-sectional emergence patterns (**B**). Increasing age was associated with global expansion (red) in inferior-anterior regions and with compression (blue) in posterior-superior regions (**A**). These effects began with inferior-anterior expansion in the youngest age comparison (Group 2 compared to 1), and intensified with increasing age groups (**B**). Compression first emerged in superior-posterior regions (Group 3 compared to 1), and later affected the middle of the brain in older age groups. (**C**) 3D representation of the continuous age results shown in (**A**), and corresponds to the Supplemental Video 1 file which shows a rotating view of these effects. (**D**) Expansion (red) and compression (blue) between the outer edges of the brain plotted as a function of continuous age. (**E**) Expansion (red) and compression (blue) effect sizes compared across age groups demonstrates the earlier emergence and stronger effect of expansion compared to compression.

4. The two datasets could not be used for a replication of all the investigated effects, due to differences in sample characteristics and neuropsychological tests applied. While I agree that a replication in a

sample with somewhat different settings can speak to the robustness of the method and results, a more direct replication particularly of the cognitive effects, but also the clinical status effects would be valuable. This could be achieved by either including yet another dataset which has the relevant data or via an internal validation of the results, i.e. either via a sample split of OASIS in discovery and test or via a, e.g., bootstrap approach testing for stability of effects with random left-out parts of the sample. I think this would further strengthen the conclusions of the manuscript.

We agree that there are differences across datasets, and that internal validation within OASIS is a complementary approach to external replication. At the suggestion of the reviewer, we have now performed internal validation with a bootstrap approach for all analyses performed in the OASIS dataset using the “boot” R package and 1000 iterations of random sample with replacement. Using this approach, we demonstrate that our original results did not demonstrate significant bias, providing support for the observed results. This has now been added as a sensitivity analyses (see the new “Sensitivity Analyses” Results section) and provided as **Supplementary Figure 5** replicated below:

To test internal validity of our analyses, we performed bootstrap analysis with replacement for all primary analyses (global distance and homologues for age, clinical status, and cognitive domains). Bootstrapped results demonstrated no significant bias in overestimation of the observed results, as the observed effect size fell within the 95% confidence interval (see Supplementary Figure 5).

A Global Distance

B Regional Homologues

Supplementary Figure 5. Sensitivity analyses demonstrating internal validation of major results. To test internal validity of our analyses, we performed bootstrap analysis with replacement for all primary analyses within (A) global distance and (B) regional homologues (age, clinical status, and cognitive domains). Histograms representing bootstrapped results demonstrated no significant bias in overestimation of the observed results, as the observed effect size (red solid line) fell within the 95% confidence interval (orange dotted line).

5. In the supplementary figures 1 & 2, effect sizes are reported, but in figure 1 with the label 'z ratio' on the x-axis, in figure 2 with the label 'effect size'. This should be made consistent and explicit – what is meant by 'effect size' in figure 2 exactly, how was it measured? And how is a z ratio reflecting an effect size (as in figure 1)?

Thank you for pointing out that these terms were not clear. The “z-ratio” is a statistic representing the effect size and is standard output from the “emmeans” package in R (beta value divided by standard error). It is similar to a “t” statistic, however it contrasts the observed statistic against a normal distribution rather than against a t-distribution (which has a slightly different shape from a normal distribution). The z-ratio can be interpreted as you would a t-statistic, because the t-distribution approaches normality as the sample size increases (in large samples, as ours is).

This explanation has now been added to the Methods for clarity:

All effect sizes are reported as “z ratios”, which is a statistic representing the effect size and is standard output from the “emmeans” package in R (beta value divided by standard error). The z-ratio can be interpreted as you would a t-statistic, however it contrasts the observed statistic against a normal distribution rather than against a t-distribution, which has a slightly different shape from a normal distribution.

We have now updated **Supplementary Figure 2** to use the term “z ratio” to more accurately represent the statistic and be consistent with the rest of the figures that use the “z ratio” terminology.

Reviewer #3 (Remarks to the Author):

Thank you for the opportunity to review the manuscript of Escalante and colleagues. The study examines age-related neuroanatomical changes as linked geometric patterns. Accordingly, the study quantified age-related neurostructural changes by measuring global expansion and compression, as well as distances between homologous regions in structural MRIs from 1059 individuals from the OASIS study (discovery sample) and 642 individuals from a replication sample (CamCAN). The study found that aging was associated with global shape expansion in inferior-anterior gradients, compression in superior-posterior gradients, and regional expansion between frontotemporal homologues. These patterns were linked to clinical impairment and cognitive deficits

The study has the following strengths

- 1. The issue of how best to capture age-related brain changes is topical and currently popular*
- 2. Substantial sample size of the discovery (OASIS) and independent validation datasets (CamCAN)*
- 3. Open science framework – i.e., use of public dataset – code availability for review*
- 4. Age range of the sample covers most of middle and late adulthood*
- 5. Use of established pre-processing pipelines as per the human connectome project*
- 6. Innovative approach to the study of the effect of age on the spatial patterns of neuroanatomy.*
- 7. Examination of the functional significance of the neuroimaging metrics*

We thank the reviewer for recognizing the strengths of our study.

Areas for revision/clarification

1. The OASIS sample includes longitudinal data – could the authors consider a pilot analysis to test whether the population level age-related changes also reflect intra-individual trajectories?

We refer the reviewer to the response to Reviewer 1, Comment 1, which also suggested we perform a longitudinal analysis. We believe our longitudinal results, although restricted by a reduced sample and statistical power, as well as a much shorter follow-up time (years rather than decades) provides converging evidence for the patterns of expansion and contraction we observed.

2. Global distances: please provide some justification for the number of points used – was it random? How does this number affect results

We refer the reviewer to the response to Reviewer 1, Comment 2, which also asked about the number of points. We have now shown that our results replicate with fewer points (15 locations, 225 points) and more points (30 locations, 900 points), providing robustness to our results with varying methodological parameters.

3. Global distances: The way this is explained in the method is that the axis considered for expansion-compression was along the sagittal plane (i.e., distances were calculated between a point on the cortical ribbon of the left hemisphere and the homologous point on the right hemisphere). Please justify and if this comment is incorrect please expand and clarify.

We thank the reviewer for highlighting this crucial aspect of our methods. The reviewer is largely correct in stating that distances were calculated between points on the cortical ribbon in the left and right hemispheres. However, points were placed in the axial (horizontal) plane rather than the sagittal plane (See Figure 1). Inferences about the pattern of expansion/compression were based on the results displayed in Figure 2, where expansion was observed in inferior/anterior gradients, and compression in more superior/posterior gradients. We hope to have clarified this issue for the reviewer and we have updated our text to improve clarity of this crucial aspect of our analyses.

Update to the Results section:

*This was achieved by calculating global distances between equally spaced points on the outer cortical surface of the brain ($n = 400$), equally spaced along inferior to superior and anterior to posterior directions **on the axial plane** on the T1-weighted MRI (see **Methods; Figure 1A**). The **Euclidean distance** between each **cross-hemisphere** pair of points was quantified.*

Updates to the Methods section:

*Specifically, we placed 400 points on the outer cortical surface **on the axial (horizontal) plane** such that they were equally spaced along 20 locations in the inferior-superior (z) and 20 locations in the anterior-posterior (y) direction.*

4. The choices for the basic model require further clarification – why no sex-specific analysis instead of modeling sex? Or perhaps both, the clarify the role of sex as it is a major moderator of brain aging

While we agree with the reviewer that sex differences are a critical factor to examine, it was not the primary goal of the current study. We included sex as a covariate in all of our models to account for the potential main effect of sex differences in head size, and therefore, distances between points or brain regions. We did in fact find a main effect of sex, which we now report in the manuscript to increase transparency, shown below:

Interestingly, covariates of no interest such as total intracranial volume and sex did in fact demonstrate significant main effects on distance gradients (intracranial volume: $F(1978.16) = 774.84$, $p < 0.0001$; sex: $F(1977.41) = 73.99$, $p < 0.0001$), however, further investigation of these factors were beyond the scope of the primary research question.

In order to properly investigate the interaction between sex and distances, we would need to include a sex by distance interaction term in our models to see if sex differentially impacts the gradients of expansion or compression. This is a more nuanced question, as we did not have any a priori hypothesis about how particular gradients or spatial relationships between regions would be differentially impacted by sex. In addition, including the interaction term would decrease the power of our statistical models. We have chosen to save the thorough, thoughtful investigation of sex differences for a subsequent follow-up study that will do this important topic justice.

5. On the same topic, why the inclusion of the total intracranial volume – could the authors produce any evidence that this is correlated with distances between points? This is intriguing as the association between volume and geometry at least in regional brain anatomy is not widely examined or known.

We included total intracranial volume (TIV) in our analyses to control for the potential of overall head/brain size to contribute to variability in distance measures. This covariate helps to ensure that we are assessing the relative, rather than the total, distance that is expanding or compressing. TIV is commonly used as a covariate in studies of regional atrophy to account for differences in head sizes which may be exacerbated by sex differences.

In our models, TIV has a significant main effect on distance. This was not a primary variable of interest in our initial analyses, however, we agree with the Reviewer that this is an interesting and not-well studied association. Therefore, we have now added this information into our results section, reproduced below:

Interestingly, covariates of no interest such as total intracranial volume and sex did in fact demonstrate significant main effects on distance gradients (intracranial volume: $F(1978.16) = 774.84$, $p < 0.0001$; sex: $F(1977.41) = 73.99$, $p < 0.0001$), however, further investigation of these factors were beyond the scope of the primary research question.

6. Could the authors comment how their data compare with other global measures of aging such as brain-age-gap-estimation? What aspect of the expansion/compression is associated with other biological meaningful measures of aging?

Comparing our results with brain-age-gap estimations and other measures of brain age is a very good suggestion. We have now added this comparison to the Discussion, reproduced below:

For example, creation of reference curves of global compression and expansion patterns associated with “normal aging”, such as done with Brain Age Gap Estimation models^{38,40}, could be created from much larger datasets and used to identify individual participant’s patterns of compression or expansion that may be related to known neurodegenerative disorders or clinical impairment.

In regards to other biological meaningful measures of aging, we plan to thoroughly address this question in future work, especially in datasets that have concurrent measures of pathology and/or functional measures that we can tie to the current work. This is further elaborated on in the future directions section of the Discussion:

Therefore, future directions include the examination of how features such as white matter structural connections between regions^{41,42}, regional microstructural properties such as demyelination⁴³, vascular integrity^{44,45}, synaptic loss^{46,47}, and other molecular contributors that affect structural scaffolding in the brain may contribute to our findings. Further, an investigation on how age-related geometric differences in shape affect functional dynamics^{21,48}, such as neural oscillations and functional connectivity, may provide more insight on how these processes lead to age-related cognitive decline in older adults.

Reviewer #3 (Remarks on code availability):

Looking at the script, I would have liked to see more detailed instructions

All code needed to reproduce these results and figures are now included in the Code Ocean facility, including README files.

Response to Remaining Critiques

Below, we respond to each of Reviewer 1's new comments in a point-by-point manner. We have included new analyses and figures/tables that demonstrate that our results are again robust to these concerns.

Response to Reviewer 1:

1. In relation to my Comment 1c, I meant that a point on the left hemisphere could be on a gyrus whilst the corresponding point on the right hemisphere could be on a sulcus or vice versa. Distances could significantly differ when measured from gyrus-to-gyrus, gyrus-to-sulcus, or sulcus-to-sulcus. Hence, on cross-sectional data like the one used in the study, for the same point of interest (e.g., point at the far end of anterior-inferior axis), a hypothetical scenario of one subject having a gyrus-to-gyrus distance vs another with a sulcus-to-sulcus distance will lead to a biased conclusion of compression. Reducing the number of points from 20 to 10 does not resolve this. Hence, either show that this effect is relatively minor and doesn't change the conclusions of the study or highlight potential specific effects of between-subject folding variations in the Discussion.

We thank the reviewer for further explaining this point which was not entirely clear to us previously. We have addressed this issue by classifying each computed point on the cortical ribbon in the left and right hemispheres as belonging to a gyrus or a sulcus according to the Destrieux Atlas (Destrieux et al., 2010). We were then able to classify each measured distance as either gyrus to gyrus, sulcus to sulcus or mixed (gyrus to sulcus or vice versa). In examining the distribution of these types of measured distances, we observed that 88% of distances were measured gyrus to gyrus, 1% sulcus to sulcus and around 11% mixed. In other words, the majority of our global distance measurements came from gyri and a much smaller fraction were between sulci.

To verify that our findings are robust against this potential difference related to sulcus to sulcus distances or mixed distances, we reanalysed our data using only those points that were measured from gyri to gyri and compared the results to those previously observed. In addition, we computed the correlation between the spatial maps in the original and restricted datasets (using the estimated model coefficients). We used the exact same model formula as described in the manuscript:

Distance ~ Sex + Euler + eTIV + Patient status + Yposition * Zposition * Age + rand(MR session).

Here are the results in Table form:

Supplementary Table 2. Comparison of results of exclusively gyri to gyri points vs. original results.

Variable	Restricted Data	Original Data	Correlation Maps
Continuous Age ¹	F=2.34, p < 0.0001	F=3.04, p < 0.0001	0.965
Age Group ¹	F=1.14, p < 0.02	F=1.17, p < 0.005	0.968
Fluid Intelligence	F=1.12, p < 0.05	F=1.71, p < 0.0001	0.961

In reference to analyzing the CamCAN data while only including the points that measured gyri to gyri (88% of the data points) compared to original analysis that includes all of the measurement points (i.e., gyri to gyri, gyri to sulcus, sulcus to sulcus).

We also show here the Figures obtained with the restricted gyri-to-gyri dataset (compare with Supplementary Figure 7).

Continuous Age:

Age Groups:

Fluid Intelligence:

In short, these analyses show that **there is relatively little impact of the type of distance measure (e.g., gyrus to gyrus, sulcus to sulcus) on our observed results**. Most of the measurements are between points on gyri and when measurements between sulci and those of the mixed type are removed from the data, the results do not substantially change. We therefore conclude that it is unlikely that differences in the type of measured distances between participants are biasing our results. However, these new results have been added to the “Sensitivity Analyses” section of the manuscript, with added text shown below:

To ensure our global expansion and compression results were not driven by the influence of gyrus to sulcus or sulcus to sulcus point pairs, we restricted our analyses to confirmed gyrus to gyrus pairs according to the Destrieux Atlas. Gyrus to gyrus pairs accounted for 88% of observed distance measurements, while a minority of measurements

was represented by gyrus to sulcus (11%) or sulcus to sulcus (1%) point pairs. Re-analyzing our data using distance measurements restricted to the gyrus to gyrus pairs demonstrated very consistent results for the effects of continuous age ($F = 2.34, p < 0.0001$; spatial correlation with original results = 0.965), age group ($F = 1.14, p < 0.02$; spatial correlation with original results = 0.968), and fluid intelligence ($F = 1.12, p < 0.05$; spatial correlation with original results = 0.961).

2. In relation to my Comments 2 and 3, I disagree with the authors' statement in the revised text that the spatial patterns are generally the same, which is not the case especially towards the superior direction when the number of locations was reduced from 20 to 15. The fact that the differences are clearly visible brings back my concern about the robustness of the results. The authors should also ensure that robustness is not only tested by visual inspection of spatial patterns but also on all statistical analyses of associations with clinical and cognitive variables.

We appreciate the reviewer's concern regarding visible differences when reducing the number of sampled data points from 20x20 to 15x15, especially in the superior direction. Because our distance-based method for characterizing brain geometry is relatively new, we recognize the importance of thoroughly testing how various analytical choices affect the robustness of our findings.

Accordingly, we employed three strategies to assess robustness:

1. **Replication within the same dataset (bootstrapping):** We resampled within a single dataset using the same analytical procedures to evaluate the stability of our results.
2. **Replication across different datasets (OASIS and CamCAN):** We applied identical analytical methods across two distinct datasets to determine whether our findings generalized despite differences in demographics and clinical characteristics.
3. **Replication under varied analytical parameters:** Within the same dataset, we systematically varied the type of distance measure (global vs. regional centroid), experimental design (cross-sectional vs. longitudinal), the number of factors in the statistical model (ranging from 10 to 20), and the number of sampled points on the cortical ribbon (225, 400, and 900).

Overall, these analyses indicated that our core results remained largely consistent under each condition, suggesting a robust pattern. However, we acknowledge the reviewer's observations of noticeable differences, particularly in the superior region, when the sampling density was reduced from 400 (20x20) to 225 (15x15). This is not entirely unexpected given that it was not initially clear how many data points were required to accurately capture the brain's geometric shape.

To further address these discrepancies, we calculated Pearson correlations of the model coefficients across the different sampling densities. The table below shows that while the correlations between 400 and 225 sampled points are lower, those between 400 and 900 sampled points are considerably higher, indicating that 225 points may be insufficient whereas 900 does not introduce substantial additional benefit over 400.

Supplementary Table 2. Correlations when comparing results using 20x20 to 15x15 and 20x20 to 30x30.

Variable	Correlation(400,225)	Correlation(400,900)
Continuous Age ¹	0.848	0.975
Age Group ¹	0.786	0.941
Clinical Status ²	0.662	0.949
Episodic Memory ³	0.754	0.899
Executive Function ³	0.750	0.861
Working Memory ³	0.695	0.822

Results for correlations computed when examining 15x15 (225), 20x20 (400), and 30x30 (900) data points in whole brain analysis. Lower correlation values in the second column suggest that 400 data points include more detailed information about the brain than 225 data points. However, high correlation values in the third column signal suggest that increasing to 900 data points includes about the same amount of information examined when using 400 points.

These correlations confirm that sampling at 225 points introduces more variability than sampling at 400 or 900 points. We will add these qualifications and quantifications to the manuscript to clarify why 225 points may not adequately approximate cortical geometry, whereas 900 points may not offer substantial improvement over 400.

In short, whereas we agree that there are visual differences between the maps generated on the basis of data with different amounts of sampled data points, this is not unexpected and provides useful information about the minimal sampling requirements to obtain stable statistical patterns. We have updated our manuscript accordingly, and this table now appears in the manuscript as **Supplementary Table 2**. Again, we thank the reviewer for highlighting this aspect of our analyses.

Modified text in “Sensitivity Analyses” section:

*The association between global shape change and age remained significant regardless of the number of points (225 points: $F(81,324968) = 10.03, p < 0.0001$; 900 points: $F(81,1455126) = 37.72, p < 0.0001$), and spatial gradients of expansion and compression were generally consistent with the original results, especially when comparing 900 sampled points to the original parameter of 400 sampled points (**Supplementary Figure 4A**; see **Supplementary Table 3** for quantification of spatial correlation across parameters and analyses).*

3. There is also another issue with the calculation of Euclidean distance between regional homologues. The distance was measured between the center of masses, but homologous regions are not perfectly symmetric and could have differences in their y- or z-axis locations; hence, connecting a straight line will not be perfectly horizontal. This can also bias the expansion/compression conclusions because the hypotenuse of a right triangle is longer than the base. This bias could be the reason why results in Figures 2 and 4 are largely inconsistent.

We thank the reviewer for pointing out this important issue. To address this point we computed for each MR session the degree to which each homologue pair deviated from a straight line. We did this in the following way:

Deviation in Y = $\text{abs}(\text{hom1 y coordinate} - \text{hom2 y coordinate})$

Deviation in Z = $\text{abs}(\text{hom1 z coordinate} - \text{hom2 z coordinate})$,

We then calculated the total deviation by summing the deviation in Y and Z. This produced for each MR session and for each homologue pair a measure that indicates the degree to which the pair deviated from a straight line.

First, we note that the overall total deviation from a straight line was relatively minor (i.e., majority of data had deviation < 4 ; see new **Supplementary Figure 6A** below) and the distribution was relatively well-behaved (i.e., there were few outliers; see new **Supplementary Figure 6B** below).

Second, we then tested whether the degree of total deviation impacted the crucial region by age interaction that we report in our study (for details see pages 6-7 of our manuscript):

Distance \sim Sex + Euler + eTIV + volumeR1 + volumeR2 + Patient Status + TotalDeviation * Region * Age + rand(MR session),

In this model the triple interaction Total Deviation * Region * Age was not statistically significant ($p = 0.2$), suggesting that the degree to which the homologue distance deviates from a straight line does not impact our results.

In addition, restricting the data to homologue pairs whose total deviation from a straight line score was < 10 did not impact the crucial Region by Age interaction (original $F = 102.44$, $p < 0.0001$, current $F = 101.67$, $p < 0.0001$; see new **Supplementary Figure 6C** below). The correlation between the estimated regional coefficients in original and the restricted data was $r = 0.999$.

Supplementary Figure 6. Evaluation of deviation of straight line between homologous regions on results. **A**, Histogram demonstrating the overall total deviation from a straight line was relatively minor (i.e., majority of data had deviation < 4). **B**, The distribution of the deviation, shown as cumulative percentage, demonstrated few outliers. **C**, Visualization of continuous age effects when restricting analyses to homologue pairs with deviation scores of <10.

This new analysis has been added in the text of the “Sensitivity Analyses” section, with updated Methods text, and added as **Supplementary Figure 6**, as shown above.

Texted added to manuscript - Results:

One potential limitation of using Euclidean distance to measure the distance between regional homologues is that asymmetry in location may result in a non-horizontal line, which may bias results. To test this, we calculated the total deviation of the line between each pair (see Methods), observing the overall total deviation from a straight line was relatively minor with few outliers (Supplementary Figure 6A-B). Replicating our main results while including the degree of deviation for each measurement resulted in a non-significant interaction of deviation by region by age ($p = 0.2$), and restricting analyses to homologue pairs with deviation scores of <10 did not impact the crucial region by age interaction (restricted data $F = 101.67$, $p < 0.0001$; Supplementary Figure 6C; spatial correlation with original results, $r = 0.999$). Together, these control analyses suggest that the degree to which the homologue distance deviations from a straight line does not impact our results.

Texted added to manuscript - Methods:

For the analyses that relied on the classification of global distance measures as gyrus or sulcus we consulted the Destrieux atlas for obtaining the appropriate label (Freesurfer file `aparc.a2009s+aseg`). Analyses performed on the CamCAN dataset were restricted to gyrus-to-gyrus distances and relied on the same model formula as for the unrestricted data. To assess deviation from a straight line, we quantified how much homologue pairs diverged while holding the X coordinate fixed. Specifically, we computed the deviation in the Y and Z coordinates as the absolute difference between the respective coordinates of each homologue pair. The total deviation was then obtained by summing the deviations in Y and Z. This measure provided an index of the extent to which each homologue pair deviated from a straight line within each MR session. We then tested whether total deviation impacted the critical Region by Age interaction in the following model:

*Distance ~ Sex + Euler + eTIV + volumeR1 + volumeR2 + Patient Status + TotalDeviation * Region * Age + rand(MR session),*

where variables were defined as before. Finally, analyses of data whose total deviation < 10 used the same model formula as for the unrestricted data.

Finally, in regards to the claim that Figures 2 and 4 are largely inconsistent. We agree with the reviewer that no strong conclusions should be drawn on the basis of visual assessments only. However, note that Figures 2 and 4 are derived from two quite different datasets using different types of sampled data points (one using cortical ribbon distances, the other using regional centroid distances), as well as different analytical procedures (different statistical model formulas). Consequently, it is not possible to make a direct quantitative comparison between these results. However, we maintain that, to the extent that visual assessments are meaningful, the overall pattern

of how aging affects the overall geometric shape of the brain (expansion in inferior temporal areas, compression in superior areas) is present in both Figures.

Reviewer #1 (Remarks on code availability):

There is no README file explaining what the folders and scripts represent.

We are happy to upload a README file on Code Ocean. Please note that currently we are unable to access the Code Ocean capsule because it is listed as 'under review'. We did upload all our code and provided annotations to explain the major steps. The code fully reproduces all the Figures and Tables from our manuscript. As far as we understand the Code Ocean system, users should be able to reproduce our figures with the click of a button to initiate a "reproducible run".